# Reasoning-Driven Synthetic Data Generation and Evaluation

**Tim R. Davidson**[*]                                                   *research@trdavidson.com*
*EPFL*

**Benoit Seguin**
*Google*

**Enrico Bacis**
*Google*

**Cesar Ilharco**
*Google Deepmind*

**Hamza Harkous**[†]                                                   *harkous@google.com*
*Google*

**Reviewed on OpenReview:** *https://openreview.net/forum?id=NALsdGEPhB*

## Abstract

Although many AI applications of interest require specialized multi-modal models, relevant data to train such models is inherently scarce or inaccessible. Filling these gaps with human annotators is prohibitively expensive, error-prone, and time-consuming, leading model builders to increasingly consider synthetic data as a scalable alternative. However, existing synthetic data generation methods often rely on manual prompts, evolutionary algorithms, or extensive seed data from the target distribution – limiting their scalability, explainability, and control. In this paper, we introduce Simula: a novel reasoning-driven framework for data generation and evaluation. It employs a seedless, agentic approach to generate synthetic datasets at scale, allowing users to define desired dataset characteristics through an explainable and controllable process that enables fine-grained resource allocation. We show the efficacy of our approach on a variety of datasets, rigorously testing both intrinsic and downstream properties. Our work (1) offers guidelines for synthetic data mechanism design, (2) provides insights into generating and evaluating synthetic data at scale, and (3) unlocks new opportunities for developing and deploying AI in domains where data scarcity or privacy concerns are paramount.

## 1 Introduction

Data availability and access have been central to advances in artificial intelligence research. In recent years, the abundance of highly-diverse internet data enabled the development of increasingly capable generalist models (Gemini et al., 2023; OpenAI et al., 2023; Anthropic, 2024; Touvron et al., 2023). Despite these models' impressive versatility, widespread integration will require them to specialize on novel, uncommon, and privacy-sensitive applications. Unfortunately, specialized data in these areas is often intrinsically scarce or inaccessible, motivating enormous investments by frontier research labs (Paul & Tong, 2024; Wiggers, 2024; Cottier et al., 2025) and the rapid rise of dedicated "data foundries" (Liu, 2025; Vinn & Hu, 2025). However, creating specialized datasets manually is expensive, time-consuming, and error-prone (Chen et al., 2023; Gilardi et al., 2023; Hosking et al., 2024), leading many to consider synthetic data as a promising,

---

[*]Work done during an internship at Google.
[†]Correspondence to: harkous@google.com. Author contributions at the end of the paper

scalable alternative (Singh et al., 2024a; Abdin et al., 2024; Guo et al., 2025). Nevertheless, how to best balance the various desiderata of synthetic data generation at scale remains an open question.

Beyond limiting AI's progress to data-rich domains, e.g., coding and creative writing, the field's reliance on real-world data imposes significant structural, safety, and operational costs. The high cost of manual data collection creates an economic threshold that marginalizes contributions from the broader research community (Chen et al., 2023). This dependency also enforces a reactive approach to safety; models can only be hardened against rare edge cases after failures are observed in deployment, a practice untenable for safety-critical systems. Operationally, the static nature of real-world data slows down development cycles, in stark contrast to the programmable workflows that a synthetic-first approach enables. Furthermore, the "black box" nature of real data makes mitigating societal biases and untangling ownership rights intractable issues that are addressable by design when the data generation process is fully controlled (Bolukbasi et al., 2016; Buolamwini & Gebru, 2018).

Using synthetic data to solve the specialized data bottleneck faces three distinct challenges concerning (i) the definition of "good" synthetic data, (ii) designing a mechanism that meets realistic requirements, and (iii) conducting generalizable evaluations. Generally, "good" synthetic data is discussed along the axes of "*quality, diversity, and complexity*" (Havrilla et al., 2024). As a running example, suppose our goal is to generate a dataset of cat images to train a specialized classifier. Instead of indicating the usefulness of data (Swayamdipta et al., 2020; Marion et al., 2023; Tirumala et al., 2023), "quality" commonly refers to how well data points fit specific requirements. For instance, if the intention is to generate an image of a "red cat", does the resulting image have a cat in it, and, is that cat indeed red? Meanwhile "complexity" can refer to how confusing, uncommon, or elaborate a specific data point is (Ethayarajh et al., 2022; Shao et al., 2023; Fu et al., 2023) but is often equated with the relative concept of "difficulty". This, in turn, begs the question: difficult for whom and under what circumstances? In the case of our red-cat image, a complex example might be a partially obscured cat or one lying in the shadows. Finally, "diversity" offers both a global and local interpretation: does the generated data globally cover the main factors of interest, and does it locally exhibit sufficient variation of specific factors? In our case, global factors could be represented by "cat breed", "color", and "environment" and the various members of these factors. Locally, we would be interested in the number of different interpretations of a combination of factors, e.g., different takes on a red cat lying on a couch. Most work on synthetic data focuses on how to best interpret "good" data, often only optimizing for a subset of the above desiderata (Havrilla et al., 2024).

The requirements of the second synthetic data challenge are less commonly discussed, namely, what are the *mechanism* criteria for generating good data? We posit that realistic deployment conditions require a mechanism that works at scale while maintaining explainability and control. Scalability is a non-negotiable: scaling-law research has repeatedly vindicated the bitter lesson that *more* good data improves outcomes (Kaplan et al., 2020; Hoffmann et al., 2022). Consequently, this quickly disqualifies methods that rely on elaborate custom prompts (Gupta et al., 2024; Xu et al., 2024b; Yu et al., 2023). Legal and operational requirements further dictate the need for transparent "audit traces", explaining the ingredients used to generate specific data points. This is problematic for methods based on stochastic, evolutionary algorithms (Mehrotra et al., 2024; Fernando et al., 2024). As will be shown in this work, not all data problems are created equal; some benefit from more quality control and others from higher complexity or local diversity. Optimal resource allocation thus requires fine-grained control over the data generation process.

The third challenge concerns evaluation: how well does the generated data intrinsically approximate a target distribution and how effective is it to train a model for a specialized downstream use case? For the latter, the most common approach is to generate data that simulates an existing benchmark, train a model on the synthetic data, and measure the trained model's performance on the benchmark's hold-out test set. However, benchmark datasets are incomplete samples of a target domain, often contain incorrect labels (Northcutt et al., 2021), are unbalanced (van Breugel et al., 2024), "under annotated", e.g., they lack multi-level annotations and details of interest, and risk being leaked into pre-training datasets. Hence, naively optimizing the data generative process toward a benchmark dataset can be deceiving. It can lead to overfitting, where synthetic data is optimized to fit the specific quirks of a benchmark dataset, which may not be representative of the target domain. Care should thus be taken to evaluate synthetic data approaches on diverse, *recent* datasets to improve generalization and minimize potential upstream contagion.

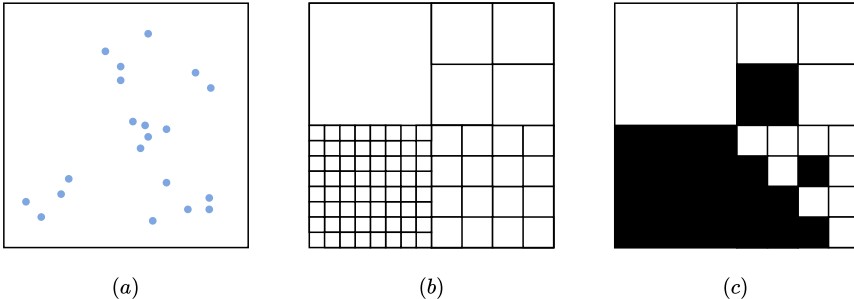

Figure 1: **Synthetic Coverage Examples.** The outer squares of (a-b-c) represent a semantic factor of interest, e.g., "*cat type*." (a) Characterizes "random" sampling behavior, with no notion of a global coverage space. This often results in samples clustered around semantic modes and misses edge cases. The grid-like structures in (b-c) represent discrete semantic spaces defined by a taxonomy's leaf nodes at increasing levels of granularity. For example, the first level could represent "*cat type*" broken down into "*domestic, big wild cats, small wild cats*, and *feral*", whereas a square at a lower level might represent a specific cat breed like the "*British shorthair*." (b) Represents perfect global planning at increasing granularity; and (c) shows global planning with progressive coverage loss, e.g., missing the branch "*big wild cats*" entirely (bottom left) or missing specific breeds (bottom right).

In this work, we propose Simula, a framework for synthetic data generation and evaluation that prioritizes transparency, fine-grained control, and scalability. Unlike many existing efforts that rely on opaque, seed-dependent processes, Simula employs a "reasoning-first" methodology, constructing each data point from first principles. This approach ensures that Simula's generation capabilities improve naturally with the reasoning capabilities of its underlying models, making the system inherently future-proof. Our three-stage pipeline first maps the conceptual space of the target domain to ensure coverage, then uses iterative agentic refinements to populate it with diverse and complex examples, and finally uses a dual-critic filtering stage to enhance quality. We rigorously test the core reasoning assumptions underlying our approach and demonstrate its efficacy on a series of carefully designed experiments using a variety of datasets. Our contributions are as follows: (1) a new reasoning-first framework for synthetic data generation capable of creating explainable and controllable data at scale; (2) a comprehensive empirical analysis connecting intrinsic properties of generated data (e.g., complexity, diversity) to extrinsic downstream model performance, offering novel, actionable evaluation metrics and insights into scaling synthetic data; (3) a set of pragmatic principles that establish mechanism design – how data is generated – as a distinct research axis, orthogonal to the traditional focus on what constitutes "good data."

## 2 Simula: A Reasoning-First Framework for Data Generation and Evaluation

Suppose we want to create a dataset with the description $y :=$ *"A dataset of stories about cats."* Due to the under-specification of $y$, it is infeasible to exhaustively describe the space of all datasets $\mathcal{Y}$, that fit this description. This is problematic as it prevents us from developing an actionable notion of coverage, i.e., given a dataset $\mathcal{D}_y \sim \mathcal{Y}$, what area of $\mathcal{Y}$ does it represent?

### 2.1 Using Taxonomies to Capture Dataset Coverage

To formulate a first-order approximation of $\mathcal{Y}$, we start by disentangling our target dataset into its prime factors of variation.[1] For example, a dataset that fits description $y$ might consist of data points covering "*cat type*," "*story format*," and "*intended audience*." In Simula, a multi-modal model (M3) is used to propose factors $f_i$ based on a set of human-provided instructions, e.g., a description like $y$, and/or a sample of existing data $\mathcal{S}$. These factors can be accepted or rejected by an M3 (or a Human). Next, an M3 is used to expand each $f_i$ breadth-first into taxonomies, $\mathcal{T}_i$, of a (user-) specified depth $d_i$:

$$\text{M3}(y, \mathcal{S}, (d_0, f_0), \cdots, (d_K, f_K)) = \{\mathcal{T}_i\}_{i=0}^{K} = \mathcal{T}^y \tag{1}$$

---

[1]Note that perfect disentanglement is of course not always possible ([Locatello et al., 2019](#)).

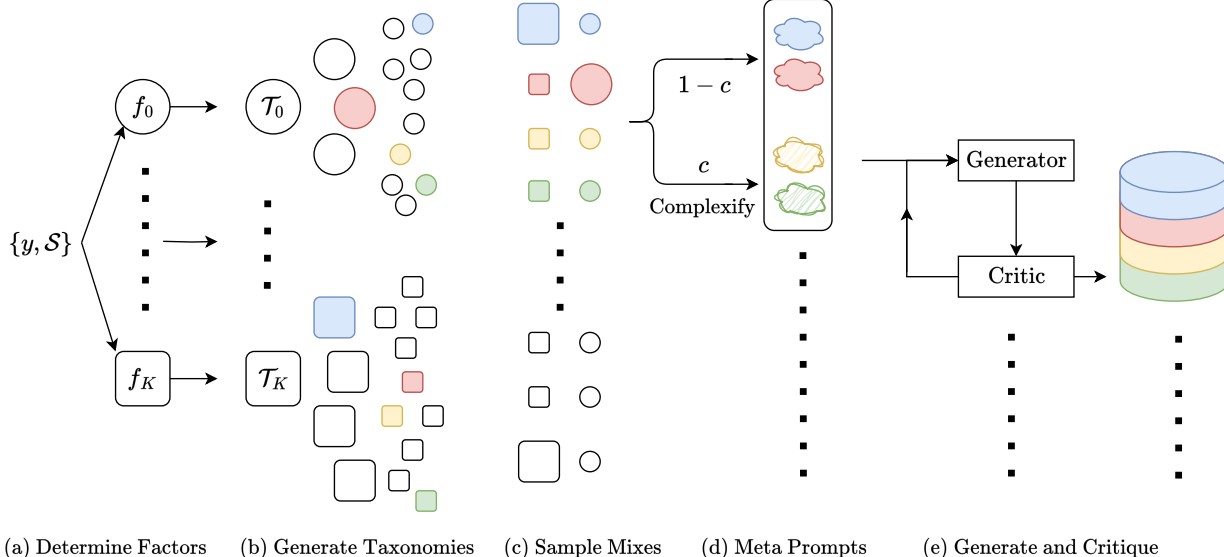

(a) Determine Factors    (b) Generate Taxonomies    (c) Sample Mixes    (d) Meta Prompts    (e) Generate and Critique

Figure 2: **Schematic of Simula Framework.** Given user instructions $y$ and/or a data sample $\mathcal{S}$, we first (a) determine factors of interest $f_i$, which (b) are expanded into taxonomies $\mathcal{T}_i$. Next, (c) nodes of $\mathcal{T}_i$ are sampled to obtain mixes, and (d) turned into "meta prompts". A user-defined fraction, $c$, of meta prompts is "complexified." (e) Finally meta prompts are used to generate data proposals by a `Generator`, and iteratively refined using a `Critic` step.

A taxonomy is a hierarchical tree structure where the root node represents a broad factor of variation, and child nodes represent increasingly specific sub-categories or instantiations. For instance, "*cat type*" can be broken down into "*domestic*" → "*shorthair*" → "*British shorthair.*" Taxonomies thus serve as a structured map of a concept space, breaking down abstract factors into concrete, sample-able attributes. This provides granular explainability and control of $\mathcal{Y}$ compared to random sampling (Figure 1.a). Intuitively, as we increase the number of factors and taxonomy depths, we sharpen our coverage control (Figure 1.b). However, this granularity comes at a potential cost: with every taxonomy expansion we risk "missing" nodes of interest (e.g., the "*shorthair*" branch), resulting in the progressive coverage loss depicted in Figure 1.c.

To mitigate potential coverage loss resulting from missing nodes, we generate $\mathcal{T}_i$ by alternating between three steps (full algorithm in App. B.4): (1) Given a node, its ancestors and its siblings, an M3 is prompted $N$ times to propose an initial set of children nodes. This sampling strategy is inspired by the "Best-of-N" literature to increase the proposal distribution and cover edge cases (Brown et al., 2024). (2) In a separate call, an M3 is prompted to act as a critic, refining the initial proposals by adding, removing, merging, or editing nodes to improve their completeness, soundness, and specificity, leveraging the generator-critic gap (Huang et al., 2024). Optionally, (3) after generating all nodes of a specific level, an M3 is prompted to generate a "plan" for the next level. This enables consistent and fast parallel generation by ensuring a similar degree of granularity at different node expansions on the same level across independent predictions. At each step, the M3 also has access to the user-provided input $y$, and/or a sample $\mathcal{S}$ from the target distribution.

## 2.2 Controlled, Explainable, and Scalable Data Synthesis with Agentic Steps

Having obtained the taxonomies $\mathcal{T}_i$, we can generate synthetic datasets that fit our requirements through two phases: taxonomic sampling (Figure 2.c) and agentic refinement (Figure 2.d-e). Full algorithm is in App. C.

**Sampling Strategies to Control Global Diversity.** An M3 first formulates a plan composed of sampling strategies. A strategy defines which taxonomies are sampled together and with what weight, preventing illogical combinations by grouping compatible subsets. For instance, a "children's content" strategy might combine taxonomies for simple topics and age-appropriate formats while excluding those for mature themes, thus preventing the generation of a horror novel for toddlers. By weighting different strategies, the framework can control the proportion of generated data for each target group. Given a strategy, the framework

samples nodes from the corresponding taxonomies $\mathcal{T}_i$. These sampled nodes can be thought of as data point "requirements," that, along with the original dataset instructions $y$, then guide an M3 to construct one or more "meta prompts". For instance, M3($y$, {*house cat, poem, travel enthusiast*}) becomes "*Compose an exciting haiku about a house cat who goes on an adventure.*" Finally, these meta prompts are passed to an M3 to generate data output proposals.

**Optimizing Local Diversity and Complexity.** Suppose we want to construct a dataset of size $N = 100$, and our factor and strategy selection has yielded $V = 200$ unique node-sets. Since $N < V$, our sampling budget allows for at most 100 unique node-sets with a single meta prompt each, resulting in a global coverage ratio of $100/200 = 0.5$. Conversely, for $N > V$, e.g., $N = 800$, we can sample up to four meta prompts for each node-set and cover the entire space. As the number of meta prompts per node-set grows, we gradually increase *local* diversity. However, for larger values of $N/V$, independently generating meta prompts from fixed requirements can lead to mode collapse, i.e., increasingly similar meta prompts. This is mitigated by generating multiple meta prompts simultaneously, then sub-sampling the required fraction. Next, we perform *complexification* on a fraction of the samples, by prompting the M3 to increase the complexity of the generated meta prompts and outputs while maintaining our other requirements. Optimizing local diversity and complexity this way works well for smaller sample sizes, but degrades as $N/V$ grows very large. Instead, for larger ratios, Simula can be configured to sequentially generate more diverse or complex meta prompts with previous attempts in context, allowing an M3 to reflect on previous generations.

**Enhancing Sample Quality with Critics.** The system performs a series of agentic refinement steps to optimize sample output quality. It starts with point-wise checks to ensure the generated samples pass the specified semantic and syntactic requirements coming from the taxonomy nodes. This involves letting the M3 "critique" the generated samples by providing the meta prompts used for generation and requesting an explanation and a binary verdict. Given the generated sample for the adventurous house-cat haiku above, the M3 would check if the cat in the story is indeed a house cat, if the output is a haiku, and if adventures were had. For tasks requiring outputs with a defined notion of correctness (e.g., classification or multiple-choice questions), the system employs an additional "*double-critic*" step, which independently assesses correctness and incorrectness to mitigate sycophancy bias (Sharma et al., 2024) (More on this in Section 3.1). Following these "*critic refinement*" steps, if the M3 responds with a negative verdict, the system either rejects the sample or applies automated modifications based on the explanation and repeats the critique step.

## 2.3 Evaluating and Comparing Dataset Properties

The previous section described how taxonomies and agentic steps can be combined to generate synthetic datasets optimized for diversity, complexity, and quality. Beyond generation, we are interested in evaluating the properties of datasets – both synthetic or real – and comparing them. We therefore describe two approaches that use agentic reasoning and taxonomies to evaluate and compare complexity and diversity.

**Reasoning about Complexity using Calibrated Attribute Scoring.** Different levels of data complexity are desired depending on the target task. Ideally, one would like to assign "complexity scores" to individual data points, enabling control over sample complexity as well as comparisons between synthetic and real data. This presents several challenges: (i) synthetic data generation is unsupervised, (ii) real data is generally not annotated for complexity, and (iii) "complexity" is a relative concept that depends on the domain and task at hand. To overcome these issues we propose a "batch-wise" evaluation approach based on reasoning: Firstly, batches of data points are sampled such that each data point appears $K$ times. Secondly, an M3 assigns scores to each sample in each batch reflecting their *complexity* using the dataset description $y$ (Algorithm in App. E.3). By providing a batch to score instead of individual points, the model can calibrate its outputs among the different points to reduce noise resulting from per-sample overconfidence (Zheng et al., 2023; Xiong et al., 2024). We further improve our scoring signal by breaking batch-specific score assignments into pairwise comparisons and computing Elo scores (Elo & Sloan, 1978). The resulting Elo scores enable complexity comparisons of data points across different datasets and can be used to control a sample's complexity.

**Using Taxonomies to Curate Dataset Coverage.** The importance of data selection during both M3 training and inference time is emphasized by a growing body of research (Swayamdipta et al., 2020; Marion et al., 2023; Ye et al., 2024; Xia et al., 2024; Hu et al., 2024; Hübotter et al., 2024), *inter alia.* Access to training data that accurately reflects test-time conditions and minimizes biases is essential to assess model performance, fairness, and preparedness. Despite their vital role, most datasets are sparsely labeled using high-level semantic descriptions, e.g., a "*math*" question, a "*harmful*" comment, a "*cat*" image, etc., or not labeled at all. This complicates efforts to curate optimal corpora, allocate resources (Qian et al., 2024), and catch potential misalignment between train and test sets (van Breugel et al., 2024). Using taxonomies offers a way forward not only for generating synthetic data, but also for better understanding existing data. Given a dataset, we can generate or use existing taxonomies that define the conceptual space of interest. We then query an M3 to perform taxonomy assignments for each data point by prompting it with the full taxonomy in context, linking each item to the most relevant node. This process yields a detailed map of the dataset's composition, from which various coverage metrics can be derived. In our evaluation, for instance, we use a "Level Ratio Coverage" metric, which calculates the proportion of unique nodes covered by a dataset at each taxonomy level. This provides a fine-grained, actionable view into a dataset's coverage, highlighting both well-represented areas and potential gaps to fill.

## 3 Experimental Setup

### 3.1 Verifying Core Reasoning Components

Simula primarily relies on three core assumptions about M3 reasoning capabilities: they (i) can generate high-quality taxonomies; (ii) function as effective "critics" of their own outputs; and (iii) distinguish between the complexity of examples. We evaluate these using Gemini 2.5 Flash (non-thinking) (Gemini et al., 2024).

**Can M3s Generate High-quality Taxonomies?** Evaluating the quality of taxonomies is inherently challenging due to the lack of standardized criteria and methods (Szopinski et al., 2020; Kaplan et al., 2022; Kundisch et al., 2021). We differentiate between *grounded* taxonomies, e.g., phylogenetic trees, and *conceptual* ones, e.g., types of harmfulness. Given an expert taxonomy, $\mathcal{T}_E$, we compare to an M3-generated taxonomy for the same topic, $\mathcal{T}_{\text{M3}}$. Structurally, we care about completeness (does $\mathcal{T}_{\text{M3}}$ cover $\mathcal{T}_E$?), soundness (does $\mathcal{T}_{\text{M3}}$ contain irrelevant or unnecessary nodes?), and novelty (does $\mathcal{T}_{\text{M3}}$ contain relevant nodes *not* in $\mathcal{T}_E$?). We evaluate both the Simula generator-critic approach and 0-shot expansion on six real-world taxonomies. Additional experimental details are available in Appendix B.

**Are M3s Effective Critics?** We test the ability of our "double critic"—which independently queries if an answer is *correct* and *incorrect*—to verify answer correctness as follows: In the *Controlled Setting*, we take reference tasks with correct answers (and if applicable, explanations), $\mathcal{D}_{\text{true}} = \{(x_i, y_i)\}_{i=0}^{N}$, and create a corrupted copy $\mathcal{D}_{\text{corrupt}}$ by prompting an M3 to subtly change $y_i$ while keeping $x_i$ fixed. After performing this causal intervention on answer correctness (Pearl, 1994), we evaluate the double critic on both datasets. Note that, for effective critic-rejection sampling, we need the likelihood of accepting a correct answer to be high and accepting a corrupted answer to be low. In the *Empirical Setting*, we test if critic capabilities *transport* (Pearl & Bareinboim, 2011) to the model's own outputs. We prompt an M3 to generate reasoning traces and answers for the benchmark tasks covering free-form math problems (MATH, Hendrycks et al. (2021)) and multiple-choice questions in selected languages (Global MMLU, Singh et al. (2024b)). We apply double-critic rejection sampling, measuring the rejection rate and the empirical change in accuracy. We note that this differs from the full Simula pipeline where the M3 generates *both* questions and answers. The net impact of the critic in this fully synthetic scenario is tested through our downstream evaluations (Sec. 4.3).

**Can M3s Distinguish Data Complexity?** We first investigate whether complexity rankings computed using our calibrated scoring method align with human-provided complexity labels. Second, we examine whether the model's complexity scores align with its own generation and verification capabilities, i.e., we study whether its performance declines on problems it rates as more complex. For this analysis, we again use the MATH dataset, which contains explicit human-annotated complexity ratings (1-5), and selected Global MMLU subjects, where education levels (elementary, high school, and college) serve as a proxy for complexity.

### 3.2 System Versions and Datasets

**System Versions.** Table 1 shows the different Simula components evaluated. To ensure a consistent basis for comparison, we first generate fine-grained taxonomies for each task. We then evaluate system configurations by ablating different components: For the **Baseline** configuration, top-level nodes are sampled from the relevant taxonomies; each mix of nodes is then turned into a data point by the M3. The **Local** component also uses top-level nodes but incorporates agentic refinement by converting them to "meta prompts" and applying "complexification" to a fraction $c = 0.5$ of these. The **Global** component leverages the full taxonomic depth by sampling from nodes at all levels, but omits meta-prompting. Finally, we evaluate a version combining both **Local + Global** diversification, and another that adds a final **Critiquing** step to enhance quality.

Table 1: **System Versions.** We evaluate the efficacy and additive properties of different system components.

|  | Taxonomy Depth | Meta Prompting | Critic Steps |
|---|---|---|---|
| **Baseline** | 1 | × | × |
| **Local** | 1 | ✓ | × |
| **Global** | >1 | × | × |
| **Local + Global** | >1 | ✓ | × |
| **Local + Global + Critique** (i.e., Full System) | >1 | ✓ | ✓ |

**Datasets.** We chose datasets that cover a spectrum of niche, recent datasets that are not commonly optimized for, as well as common datasets whose domains are optimized for in popular benchmarks. For the former, focused on specialized tasks, we chose two datasets from the Cyber Threat Intelligence Benchmark (CTIBench) (Alam et al., 2024): **CTI-MCQ**, a multiple-choice question dataset with four choices and a single answer to assess M3s' understanding of CTI standards, threats, and mitigation (2.5k test questions); and **CTI-RCM**, a task requiring the open-ended generation of a Common Weakness Enumeration (CWE) category based on a Common Vulnerabilities and Exposures (CVE) description (1k test CVEs). We also selected a dataset from the legal domain, **LEXam** (Fan et al., 2025), which consists of 1.66k multiple-choice questions from Swiss, EU, and international law examinations in both English and German.

To cover popular domains, we selected two well-known datasets: **GSM8k** (Cobbe et al., 2021), high-quality, linguistically diverse grade-school math problems requiring multi-step reasoning (1.32k test problems); and a subset of **Global MMLU** (Singh et al., 2024b), a multilingual, multiple-choice question dataset with four choices and a single answer. To keep the data generation manageable, we selected a subset across Math, Computer Science, and Physics in English (High-resource), Korean (Mid-resource), and Nepali (Low-resource), for a total of 1.44k test questions per language. We applied a post-filtering process for the generated datasets, including de-duplication and decontamination against the real test sets (based on 13-gram overlap with a Jaccard threshold of 0.8). We ended up with 512k unique data points for each of the above datasets.

### 3.3 Intrinsic Metrics

We compute intrinsic metrics on a sample of 1k data points for each dataset and each system version.

**Embedding Diversity and Taxonomic Coverage.** Following Yu et al. (2023), we evaluate diversity by transforming datapoints into an embedding space and measuring cosine distances. For global diversity we report the average dataset-wide, pairwise cosine distance. To evaluate local diversity, we first group data points by taking the $k = 10$ nearest neighbors to each data point in embedding space to ensure semantically meaningful clusters. We then take the average pairwise cosine distance across these clusters. For both, a higher cosine distance indicates higher diversity. We compute the embeddings using the model from Lee et al. (2024b). In practice, cosine distance scores are coarse metrics that do not provide actionable insights to improve diversity. We therefore also evaluate each dataset's taxonomic coverage using the assignment-based methodology described in 2.3, which provides a more structured and actionable measure of how comprehensively data covers a conceptual domain.

**Complexity.** Our goals for intrinsic complexity evaluation are twofold: First, we want to test if the different Simula components can effectively control the complexity distribution of a synthetic dataset. Secondly, we want to compare the complexity of synthetic datasets to the "real" reference datasets. After generating synthetic datasets using the different systems, we follow the calibrated complexity scoring method outlined in Section 2.3, mixing together the various synthetic datasets with the real data.

### 3.4 Downstream Metrics

For all downstream experiments we use Gemma 3 4B as our student model (Gemma et al., 2024) and Gemini 2.5-Flash (non-thinking) as our teacher model. For datasets covering domains that are commonly optimized for such as GSM8k and Global MMLU, we use the pre-trained (PT) version as the starting point. For the more niche, recent datasets, CTI-MCQ, CTI-RCM, and LEXam, we use the instruction-tuned (IT) version. We perform 10 iterations of LoRA fine-tuning (Hu et al., 2022) with different seeds per configuration and dataset size, reporting mean accuracy for each data size and 95% confidence intervals computed as the standard error of the mean across seeds (see App. F for our hyperparameter optimization process). To isolate the impact of complexity on downstream metrics, we split datasets produced by the full system into three subsets. To avoid confounding missing complex concepts with complexification per concept, we sample by complexity score per taxonomy node. This gives a *Low Complexity* split (bottom 40% of the data), a *High Complexity* split (top 40%), and an *All Complexity* split (a random subsample of the previous two).

## 4 Results

### 4.1 Results for Core Reasoning Components

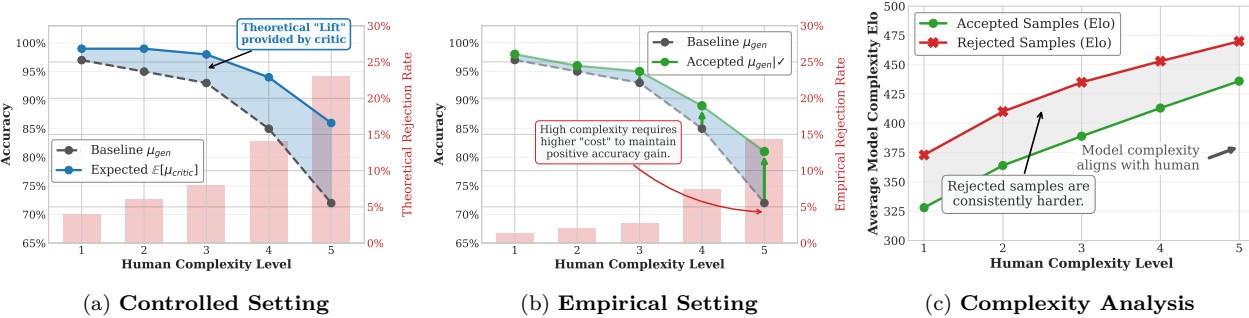

(a) **Controlled Setting**      (b) **Empirical Setting**      (c) **Complexity Analysis**

Figure 3: **Double-Critic Rejection Sampling on MATH.** (a) We establish the theoretical lift in the controlled setting, noting that rejection costs increase with task complexity. (b) Critic capabilities transport to the empirical setting but lose some effectiveness. (c) Calibrated Elo scores show model-human alignment. Stratified by complexity level, rejected samples are consistently assigned higher model complexity.

**M3s Can Generate Good Taxonomies.** Table 2 shows Simula taxonomies approximately cover those created by human experts. For conceptual taxonomies, almost all generated nodes are sound, with many novel expansions resulting in increased total coverage. Note that we are less concerned with "over" coverage, as this ensures that most edge cases are present. We can always reduce the depth of a taxonomy if we believe it to be too granular. These results support our approach of approximating a global coverage space using taxonomies. Appendix B contains further details and analyses on evaluated taxonomies, as well as limitations and examples of synthetic taxonomies.

**Double-Critic Rejection Sampling Enhances Quality.** Figure 3 summarizes our double-critic evaluation on the MATH dataset, where $\mu_{\text{gen}}$ is the model's generative baseline accuracy. In the *Controlled Setting* (Figure 3a), we measure the probabilities of the critic accepting correct, $p(y)$, and incorrect, $p(y^{\text{corrupt}})$, answers. The accepted proportion is defined as $|\mathcal{D}_{\text{accept}}| = \mu_{\text{gen}} \cdot p(y) + (1 - \mu_{\text{gen}}) \cdot p(y^{\text{corrupt}})$. The expected accuracy of these items is $\mathbb{E}[\mu_{\text{critic}}] = \mu_{\text{gen}} \cdot p(y)/|\mathcal{D}_{\text{accept}}|$. We observe a consistent theoretical "lift" ($\mathbb{E}[\mu_{\text{critic}}] > \mu_{\text{gen}}$). As human-assigned complexity increases, maintaining this lift requires a higher "cost" in rejection rate

Table 2: **Taxonomy evaluation.** Average metrics across taxonomies. Novelty and total coverage are only reported for Conceptual taxonomies, since we assume that grounded taxonomies should have no room for further extensions.

| Method | Grounded | | Conceptual | | | |
|---|---|---|---|---|---|---|
| | Completeness | Soundness | Completeness | Soundness | Novelty | Coverage |
| **Simula** | **0.74** | **0.75** | **0.78** | 0.97 | **0.94** | **1.72** |
| **0-Shot** | 0.52 | 0.70 | 0.50 | 0.97 | 0.32 | 0.83 |

$(|\mathcal{D}_{\text{reject}}| = 1 - |\mathcal{D}_{\text{accept}}|)$. In the *Empirical Setting* (Figure 3b), the critic filters the model's own outputs into accepted and rejected subsets. While still realizing a lift over the baseline, $\mu_{\text{gen}}|\checkmark > \mu_{\text{gen}}$, it is less effective. Again, higher complexity necessitates a higher empirical rejection rate to sustain accuracy gains.

**Complexity Scores Align; Rejected Samples Perceived Harder.** Figure 3c validates our calibrated scoring method: model-assigned Elo scores align with the humans ones. We also see that stratified by human-assigned complexity, rejected samples have higher Elo scores than accepted ones, indicating the critic systematically filters samples with higher perceived model complexity. Additional results are in App. D, E.

## 4.2 Intrinsic Metric Results

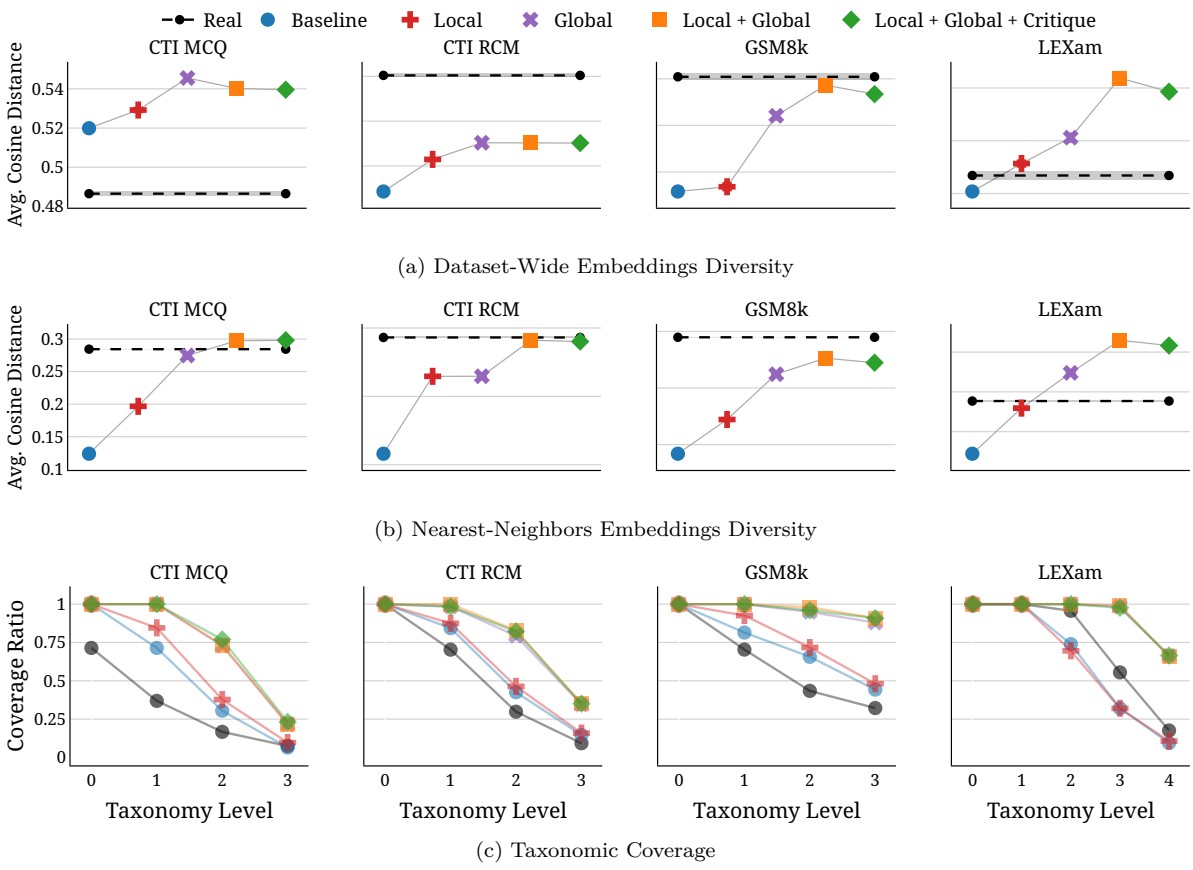

(a) Dataset-Wide Embeddings Diversity

(b) Nearest-Neighbors Embeddings Diversity

(c) Taxonomic Coverage

Figure 4: **Intrinsic Diversity Metrics.** We display dataset-wide (top) and nearest-neighbors (middle) embedding-based diversity, and taxonomic coverage (bottom). We note that Global diversification is crucial for increasing dataset-wide diversity, and that Local and Global diversification generally have an additive effect. We further note that while real data can be more or less diverse according to embedding-based metrics, it almost always covers less of the target domain than Simula variants on a taxonomy basis.

**Intrinsic Diversity.** Figure 4 shows results for the different diversity metrics. Starting with the embeddings-based diversity metrics (top and middle), we first note the Global diversification component is crucial to increase the dataset-wide embedding-based diversity (top). Second, the Local component improves over the baseline; hence meta-prompting effectively increases the diversity of the $k$ nearest points (middle). Third, the Local and Global diversification components generally have an additive effect for embedding-based metrics. While embedding-based diversity of real data can be both higher or lower than the synthetic variants, when it comes to taxonomic coverage, we observe a clear pattern: the full Simula system exhibits higher coverage as the taxonomy levels increase. Assuming our taxonomies describe comprehensive coverage of the target domain, this indicates that the real data misses large subsets of interest. Finally, critiquing does not appear to affect diversity in a significant way, almost always overlapping with the variant without critic steps.

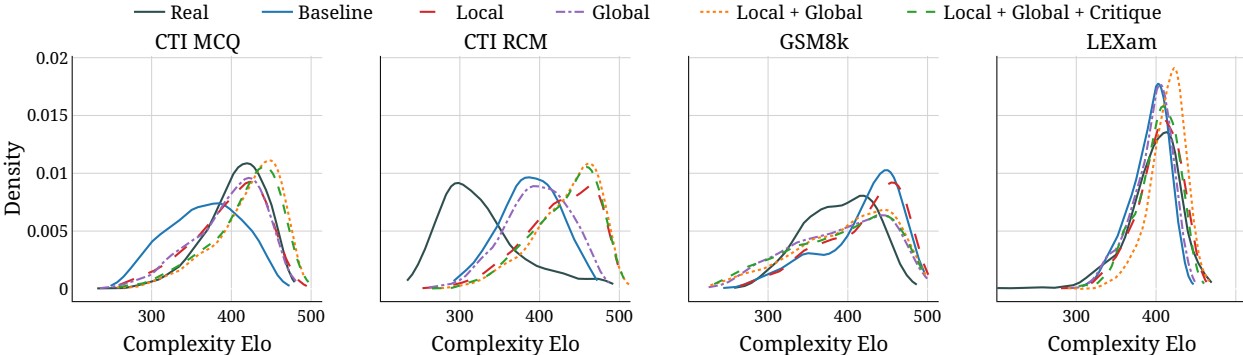

Figure 5: **Complexity Elo Distribution of Synthetic and Real Data.** We display density plots of the complexity Elo rankings for the various system versions on four datasets. We note that synthetic data can cover the entire complexity range of all datasets and that Local and Global components are generally additive, i.e., they account for different types of complexity.

**Intrinsic Complexity.** We display density plots of our calibrated complexity evaluation in Figure 5. It can be observed that the full Simula system is capable of covering the entire complexity range of real data, often even allowing for more complex data points. We also note that Local and Global components generally have an additive relation, providing different types of complexity. The different datasets have distinctly different complexity profiles, with the first three panels covering a wide range while LEXam is much more concentrated.[2] Furthermore, while increased diversity and critiquing can reduce complexity (GSM8k), as we will see in the next section, this does not necessarily have an adverse effect on downstream performance.

### 4.3 Downstream Results

In Figure 6, we show how downstream performance evolves per dataset across data sizes and system versions. Results for additional Global MMLU subsets are available in App. F.2.

**Simultaneously Optimizing all Data Axes Never Hurts.** We first observe that using the full Simula system is almost always the dominant strategy across all datasets and data sizes. This is in contrast to the Baseline, which scales the worst, indicating that the data-scaling law is a function of the data properties, not size alone. This also means that, without strong priors, using the full system is a good strategy.

**Student-Teacher Gap Impacts Scaling Laws.** We see that downstream performance of the student model at different data scales is impacted by the relative difference against the teacher's performance. In the case of CTI RCM, we see that performance saturates at around 128k, after bridging $\frac{65-40}{70-40} \simeq 83\%$ of the performance gap between the starting point of the student (40%) and the teacher (70%) model. This is not the case with GSM8k, where all versions continue to grow, as the performance of the student model (maximum of 75%) remains sufficiently far from the teacher's performance (88%) on this task.

---

[2]This aligns with the teacher model's poor performance on LEXam, as shown in Section 4.3.

**Combining Global and Local Diversity is Critical.** We see how Local and Global diversification in isolation provide suboptimal returns depending on the dataset and data size. In the case of CTI MCQ, Local diversification plateaus before variants containing a Global component. Local diversification also leads to significantly weaker performance in the small data regime for CTI RCM (4k and 16k). On the other hand, Global diversification does not surpass the Baseline in LEXam and scales slightly worse in the case of GSM8k. Combining the two improved performance on all datasets and dataset sizes.

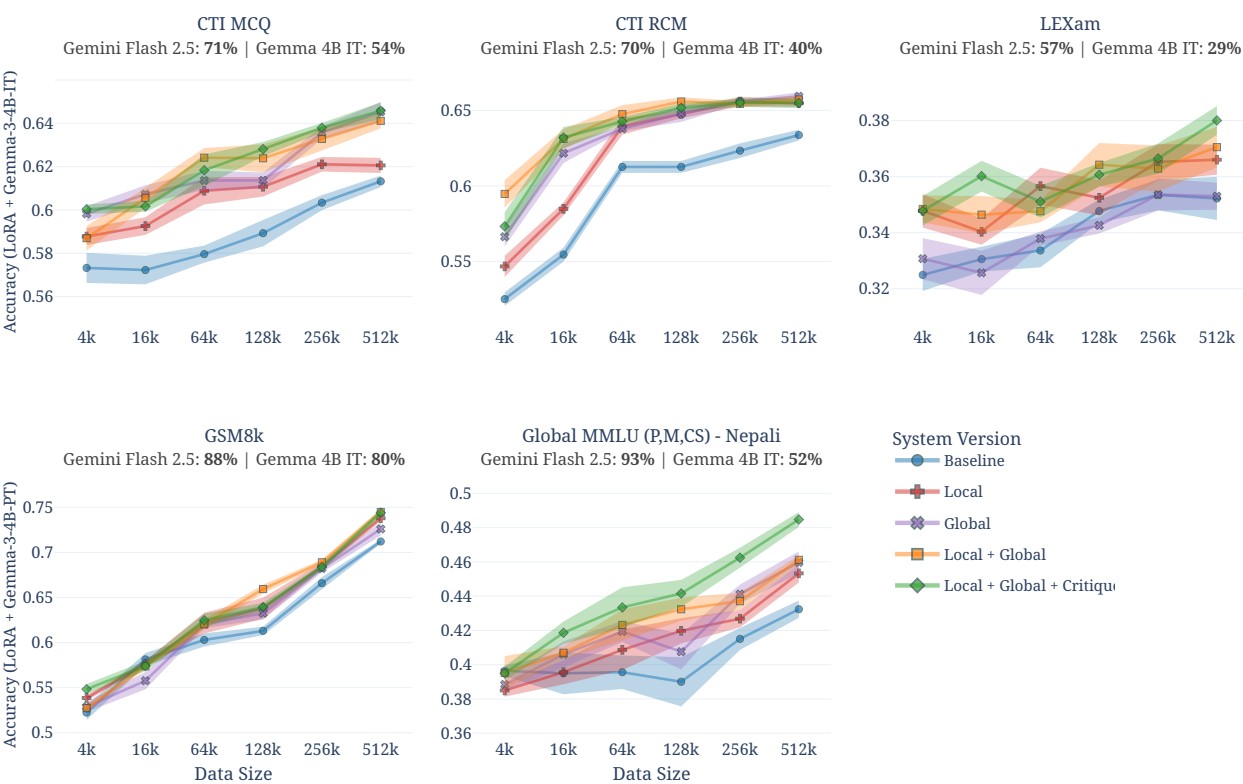

Figure 6: **Downstream Performance on Different Datasets.** We report mean accuracy with 95% CI for different system versions. We note that while the contribution of different components to performance depends on the dataset and data size, the full Simula system always reaches the best result.

**Critiquing Impact is Dataset Dependent.** Adding the critiquing step does not hurt performance on any of the studied datasets. In the case of Global MMLU it made a significant improvement, despite the rejected data being limited to only 3%. The critic rejection rate was 2% for CTI MCQ, 9% for CTI RCM, 9% for GSM8k, and 61% for LEXam. For LEXam, the high rejection rate can likely be attributed to the weaker performance of the teacher model (57% accuracy). Even when critic-based rejection sampling does not significantly improve downstream performance, we argue that adding a critic step is still desired. First, ensuring data points follow diversity requirements maintains coverage control. Second, increasing "correctness" of labels promotes causal learning, thus improving model robustness.

**Complexity is Crucial in Most Domains but Can Hurt with a Weak Teacher.** Figure 7 illustrates the impact of data complexity on mean accuracy across four datasets. The importance of complexity is most evident in the GSM8k dataset. Here, we observe a direct correlation between data complexity and accuracy, with the High Complexity split yielding a 10% accuracy gain over the Low Complexity split at 64k data items. A similar trend is present in the CTI MCQ dataset, where the Low Complexity split shows negligible accuracy scaling with increasing data size. Conversely, for the LEXam dataset, only the Low Complexity split demonstrates improved performance with scale. We hypothesize that this is a consequence of weaker teacher performance, which leads to diminished accuracy on more complex data.

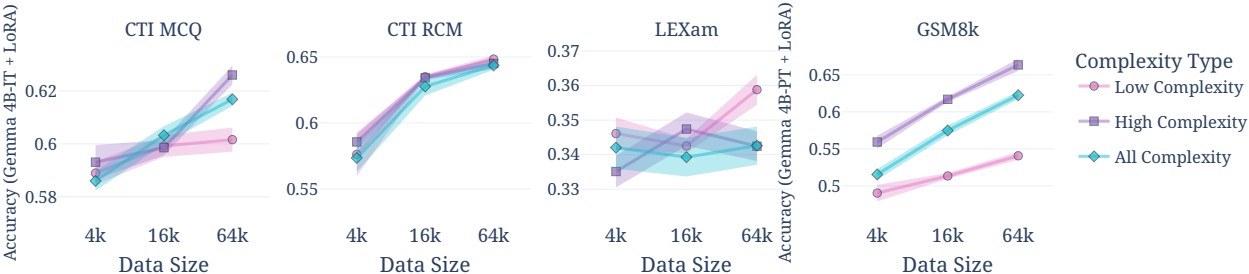

Figure 7: **Downstream Impact of Data Complexity on Mean Accuracy.** For GSM8k and CTI MCQ, higher complexity data improves performance and scaling. Conversely, for LEXam, only the Low Complexity split shows improvement, suggesting that complex data can be detrimental when the teacher model is weak.

## 5 Discussion

**Synthetic Data Generation Has No Single Optimal Solution.** "Data" is a frozen reflection of reality as it was or could be. No wonder then, that there is no single "optimal" way to generate an infinite number of possibilities. Our extensive experiments show that the impact of different data properties, e.g., complexity and diversity, depends on the target domain, the model, use case, scale, and likely many other factors. Instead of looking for silver bullets, we should thus design our synthetic data systems to be as flexible as the worlds we intend to capture. With Simula we offer a system that maintains explainability and control at scale, giving practitioners the tools to customize synthetic data to fit their unique requirements.

**Synthetic Data Evaluation is a Multi-faceted Challenge.** The evaluation of synthetic data is fundamentally challenging due to the ambiguity of its core objectives, the coarse level of existing metrics, and its disconnect from practical context. Key properties describing "good data" are ambiguously defined and inherently entangled. For instance, one could argue that covering rare domain instances falls under "diversity." However, one could just as easily attribute these to "complexity." Similarly, the challenge of generating samples faithful to many – possibly conflicting – requirements falls as much under "quality" as the other two axes. One could thus argue that an ideal evaluation target does not exist; only context-specific trade-offs. While established intrinsic evaluation metrics like embedding-based cosine distance provide a high-level signal, they fail to provide actionable insights or comparisons. By introducing alternatives like node-level taxonomic coverage and reasoning-based complexity scores, we strive to make intrinsic evaluation more pragmatic. Ironically, the very mechanisms that create a need for synthetic data – novelty, rareness, privacy – make it challenging to robustly assess any synthetic data system against canonical references. Although we evaluated on a mix of niche and popular benchmarks, availability of more high-quality, specialized references would benefit this field.

**The Life Cycle Costs of Synthetic Data.** In our experiments we compared various Simula variants to a baseline that requires up to 5x fewer inference calls per data point. Ignoring the one-off costs of creating a deeper taxonomy, this implies one could generate five times the amount of data for a fixed generation budget using the baseline system. Nevertheless, this "cheaper" larger dataset is likely more expensive when taking into account the full data life cycle. As we saw in Figure 6, data generated using Simula can be both more efficient and reach higher total performance. Since the costs of training are significantly higher than those of doing inference, smaller datasets are preferred even if they are more expensive to generate. When using synthetic data to evaluate model preparedness, larger datasets naturally require more evaluation steps.

**Limitations and Future Directions.** This work presented a comprehensive analysis of different factors relevant to synthetic data generation and evaluation. Nevertheless, our study has several limitations worth discussing. We conducted our experiments using a single model family to generate data (Gemini 2.5 Flash). As shown in Section 4.3, the teacher model used in our ablations has a significant performance gap with the student model even after finetuning, making it unlikely to be a bottleneck on downstream performance.

Additionally, as each data point in Simula is generated from first principles using reasoning-based components, we expect outputs to naturally improve with the underlying models powering the system.

To isolate the effects of different system components, we fixed the student model size and finetuning method. This approach is grounded in established scaling laws showing that performance improves with better data, regardless of model scale (Kaplan et al., 2020; Hoffmann et al., 2022). While the improvements from using more data can diminish, these reflect diminishing marginal returns due to data redundancy (Muennighoff et al., 2023) or quality (Chen et al., 2025), not a plateau in which larger models cease to benefit from more or better data. We therefore expect our findings to hold across various model scales. Following Schulman & Lab (2025), it is further reasonable to assume that LoRA finetuning approaches full finetuning performance for the evaluated data regimes. Comparing reinforcement learning to supervised finetuning on our target tasks remains an interesting experiment for future work.

Although the experiments presented here focus exclusively on text-based synthetic data, we see no reason why the same system could not be used for other modalities. We leave the exploration of these for future work.

## 6  Conclusion

Traditionally powered by data abundance, AI progress is at a junction; just as its potential is becoming evident, the specialized data needed to realize it is unlikely to be generated by humans (Villalobos et al., 2024). Synthetic data is primed to play a central role in new AI leaps – enabling the creation of specialist systems, equitable access, fairness, and preparedness evaluation. Beyond focusing on what it means to generate "good" data, this work emphasizes the importance of aligning *mechanism design* to practical requirements. With Simula, we provide a reasoning-first framework capable of generating explainable and controllable synthetic data at scale. Aided by this controllable framework, our in-depth experiments highlighted just how idiosyncratic the relation between "good" data and downstream performance is. We therefore urge the community to resist searching for "silver bullets," and instead embrace synthetic data's deep context-dependency through controllable, reasoning-driven approaches.

### Broader Impact Statement

Simula is a general-purpose framework for generating synthetic datasets. By offering fine-grained control and steerability, Simula introduces a dual-use potential. While our work is motivated by a desire to overcome latent biases and increase explainability in data generation, we recognize that these same capabilities could be leveraged by malicious actors to produce nefarious content or reinforce existing biases. For instance, the ability to precisely manipulate factors of variation could be used to generate datasets that promote harmful stereotypes or misinformation, depending on the intentions of the steering agent (M3 or human).

We highlight that Simula integrates several mechanisms designed to mitigate these risks from happening inadvertently. Firstly, it promotes transparent generation by transforming the opaque nature of real-world data into a "white box" problem. This allows for a clearer understanding and control over potential biases. Secondly, it provides auditing tools that can be used as pragmatic post-checks. The use of taxonomies enables the measurement of data coverage, offering a practical method to detect imbalanced output distributions of specific factors of interest. Similarly, our "calibrated attribute scoring" technique can be employed to identify undesirable shifts in sensitive attributes, such as assessing the level of prejudice. These inherent features aim to facilitate the detection and rectification of biases, thereby promoting responsible and ethical synthetic data generation.

## Author Contributions

**Tim R. Davidson**: Led the majority of the writing of the manuscript, including system formalization, literature review, experimental setup, and evaluation; designed and evaluated the double critic-rejection sampling approach; developed and evaluated the calibrated complexity scoring metric; introduced data coverage metrics; developed several optimizations for the taxonomy generation algorithm (e.g., best-of-N, critiquing); conducted intrinsic evaluations; provided core insights across the other parts during his internship at Google.

**Benoit Seguin**: Led engineering for Simula; implemented core components for modular, scalable data generation; developed the core library for prompting and large-scale inference used throughout the experiments; established the fine-tuning framework for downstream evaluation.

**Enrico Bacis**: Built the library for unified data representation across diverse datasets and researched suitable datasets; developed the evaluation framework for few-shot experiments and conducted these experiments.

**Cesar Ilharco**: Developed metrics for evaluating taxonomies and conducted the corresponding experiments; researched and collected suitable datasets for taxonomy evaluation.

**Hamza Harkous**: Founded and led research for Simula; designed the core system for end-to-end seedless data generation; researched and implemented the original system components, including mixing strategies, taxonomy building, meta-prompting, critiquing, and refinement; conducted downstream fine-tuning experiments for evaluation; developed the data generation setup for ablation studies.

**All authors**: Contributed to project ideation, technical discussions, and manuscript writing and revision.

## Acknowledgments

We would like to thank Nina Taft and Amanda Walker for continuous support and advice on the project and the paper. We also thank Coran Corbett for engineering contributions to Simula and Etienne Pot for supporting us in the usage of Kauldron for Gemma model training.

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

# Appendices

# A  Related Work

## A.1  Synthetic dataset evaluation.

**Early Evaluation Methods.** When evaluating a synthetic dataset, we can differentiate between *comparative* and *intrinsic* evaluation. The former expects the availability of a reference dataset. In the case of "closed" tasks that allow for narrow comparisons, we can directly compare model-generated outputs to reference solutions, e.g., single-word answers, translations, or summaries. Originally, these were done directly on the observed outputs, e.g., by comparing overlapping words or n-grams (Papineni et al., 2002; Lin, 2004; Lavie & Agarwal, 2007), *inter alia.*

**Rise of Semantic Metrics.** As tasks might have multiple different, but semantically equivalent solutions, evaluations shifted to embedding-based approaches (Patil et al., 2023). Such approaches can even be used in the absence of one-to-one reference mappings, by comparing output distributions on a dataset level (Heusel et al., 2017; Pillutla et al., 2021). Although embedding-based approaches allow for automated evaluation, they can struggle in specialized domains and miss semantic subtleties (Ethayarajh, 2019; Kashyap et al., 2023).

**Auto-Verifiable Tasks.** Then, we have a class of problems that allow for many trajectories ending in a final solution whose correctness can be automatically verified quickly, e.g., program synthesis and mathematical reasoning (Song et al., 2024), or negotiation games (Davidson et al., 2024b).

**The Challenge of Open Tasks.** What remains is a large class of "open" tasks, for which there are no well-defined criteria of correctness, e.g., creative writing, advice, and most visual media. For these, we largely rely on the reasoning capabilities of human annotators to provide quality references and to obtain pairwise preference labels (Christiano et al., 2017; Bai et al., 2022). However, creating such datasets manually is expensive, time-consuming, and error-prone (Chen et al., 2023; Gilardi et al., 2023; Hosking et al., 2024).

**Emergence of LLM Judges.** Recently, the growing capabilities of frontier models have opened up the possibility of model-based reasoning (Lee et al., 2024a; Zheng et al., 2023; Li et al., 2024a; Saunders et al., 2022; Wang et al., 2023; Madaan et al., 2024), *inter alia.* As humans and AI start to prefer AI-generated text (Zhang & Gosline, 2023; Panickssery et al., 2024), and AI prefers text by the strongest AI models (Davidson et al., 2024a), model-based evaluation is set to become increasingly prevalent.

**Intrinsic Evaluation Metrics.** In the absence of reference data, one must rely on *intrinsic* evaluation. As described in the introduction, we typically focus on the axes of quality, diversity, and complexity. When we treat quality as how well a set of data points meets stated requirements, we are quickly forced to rely on reasoning-based evaluation. Instead, if understood as the utility of a dataset for a specific downstream task, we can directly measure the lift in downstream performance, e.g., through classification accuracy. Diversity is often approximated using pairwise cosine similarity after embedding outputs into a higher dimensional space (Yu et al., 2023; Gupta et al., 2024). Without grouping the data using an appropriate clustering step, e.g., semantic clusters, average statistics are sensitive to outliers and fail to differentiate between global and local diversity. Crucially, such diversity statistics provide few semantic, actionable insights. Attempts to automate complexity scoring range from measuring the length of outputs (Shao et al., 2023), or the relative entropy over output alternatives (Ethayarajh et al., 2022; Lu et al., 2023), to generating a large solution set and using a reward model to estimate correctness (Snell et al., 2024). Reasoning-based approaches instead directly query a model to provide a "difficulty" score (Li et al., 2024a). Because models are generally poorly calibrated (Zheng et al., 2023; Tian et al., 2023; Xiong et al., 2024), and difficulty is a relative concept, such absolute scores can be noisy.

**Our Reasoning-Based Approach.** Our work continues the trend of incorporating model-based reasoning to evaluate data. Instead of using approximate statistics based on output embeddings, reasoning-based approaches provide explainable traces that can be audited and controlled. Mapping out a global coverage space using taxonomies allows end-users to quickly evaluate if their generated data meets the appropriate global diversity requirements (Sections 2.1, 2.3). We further carefully tested popular assumptions about the use of model-based critics through a series of controlled experiments. On the evaluated datasets, we find that critic-rejection sampling of synthetic outputs consistently succeeds in increasing average sample quality

(Sections 2.2, 3.1, 4.1) across different datasets and complexity levels. We also found that model-assigned complexity scores are promising proxies for human notions of difficulty, and correlate well with models' critic and generation capabilities (Sections 2.3, 4.1, and Appendix E).

### A.2 Synthetic dataset generation.

**Seed-Based Expansion Methods.** Popular synthetic data methods generally only account for a subset of the quality, diversity, and complexity axes (Havrilla et al., 2024). For example, Wang et al. (2023) start with a set of seed examples and iteratively expand them using hand-crafted semantic diversity prompts. Xu et al. (2024a) similarly expand seed examples, but focus on both diversity and complexity. The main quality check performed is to ensure that these expanded examples do not become degenerative. An attempt at increasing global diversity is done by maximizing pairwise cosine similarity through rejection sampling. The authors note that expanding semantic diversity and complexity are both positively correlated with downstream performance after fine-tuning.

**Factor Identification Approaches.** Reif et al. (2024); Chen et al. (2024); Viswanathan et al. (2024); Lu et al. (2024) use seed examples to automatically detect relevant factors through iterative sampling of the target dataset, after which factors are extracted using reasoning modules. Other approaches side-step the need for seed examples by manually inspecting or reasoning about a target dataset to find globally relevant factors of variation (Yu et al., 2023; Gupta et al., 2024; Samvelyan et al., 2024). They then sample from these factors for conditional generation. Relevant to our framework is work done by Li et al. (2024b), who attempt to generate a single, large taxonomy to cover a variety of topics. Inspired by curriculum-based learning in human education systems, the authors generate a variety of learning modules to implicitly vary complexity.

**Leveraging the Critic Gap.** Many have by now pointed out the apparent gap between current models' generative and verification capabilities (Huang et al., 2024). This gap allows models to act as critics of their own outputs (Saunders et al., 2022; Madaan et al., 2024) and has been successfully used by many of the above methods (Lee et al., 2024c; Gupta et al., 2024; Samvelyan et al., 2024; Chen et al., 2024), among others.

**Scaling Test-Time Compute.** Recent efforts in scaling test-time computation show that models are capable of generating correct outputs even for complex questions, given enough attempts (Song et al., 2024; Brown et al., 2024). Yet, how to best scale such a test-time computation budget depends on the complexity of the particular problem (Snell et al., 2024). The authors suggest that "easier" tasks most benefit from exploiting an existing output attempt through iterative refinement, whereas more "difficult" tasks benefit more from exploring a larger proposal distribution.

**Our Orchestration Approach.** With Simula, we build on many of the existing insights into the merits of optimizing quality, diversity, and complexity for downstream performance. In contrast to existing methods, we explicitly orchestrate the generative process on a dataset level to increase global control and explainability. We carefully split the data generation process into separate steps, i.e., global diversity, local diversity, complexity, and correctness, allowing end users to tailor datasets to their specific requirements (Section 2). In doing so, it becomes possible to allocate computational resources where they are most desired, e.g., by generating more meta prompts to increase the proposal distribution of complex samples or adding additional critic steps to refine the outputs.

# B  Reasoning-driven Taxonomy Generation and Evaluation

To address the inherent challenges of assessing taxonomy quality and completeness, we use a critic-model based framework for evaluation. Traditional taxonomy evaluation often relies on manual expert review, which is time-consuming, expensive, and often difficult to attain. Our proposed framework leverages the capabilities of large language models as "critic models" to provide a more automated, scalable, and reproducible evaluation.

## B.1  Defining Taxonomy Evaluation Metrics

First, a hierarchical representation for each expert taxonomy ($\mathcal{T}_E$) or model-generated taxonomy ($\mathcal{T}_{M3}$) is provided, and the critic model classifies each node into the following categories:

- **Good and Overlapping**: The node is good (well-defined, relevant to its parent node, and fits appropriately within the overall taxonomy) AND overlapping (there is a semantically equivalent node in the other taxonomy which represents the same concept).

- **Good and Exclusive**: The node is good (well-defined, relevant to its parent node, and fits appropriately within the overall taxonomy) AND NOT overlapping (there is no semantically equivalent node in the other taxonomy which represents the same concept; this concept appears uniquely in this taxonomy).

- **Redundant**: The node is a duplicate within its own taxonomy, there is another node in the same taxonomy representing the same concept.

- **Bad**: The node is irrelevant, poorly defined, misclassified, or otherwise inappropriate for its position in the taxonomy.

Based on the critic model's classifications, we compute several quantitative metrics to evaluate each taxonomy:

- **Completeness**: This metric measures the extent to which $\mathcal{T}_{M3}$ covers the concepts present in $\mathcal{T}_E$. The LM critic assesses, for each node (concept) in $\mathcal{T}_E$, whether a semantically equivalent node exists in $\mathcal{T}_{M3}$. This serves as a measure of coverage and recall, quantified by the ratio (Good and Overlapping) / (Total Good) in $\mathcal{T}_E$. Here, Total Good = Good and Overlapping + Good and Exclusive.

- **Soundness**: This metric assesses the proportion of relevant and correct nodes within $\mathcal{T}_{M3}$. The LM critic examines each node in $\mathcal{T}_{M3}$ to judge its relevance to the topic and whether it constitutes a non-redundant entry. Fewer irrelevant or incorrect nodes result in greater soundness. This serves as a measure of precision, quantified by the ratio (Total Good) / (Total Nodes) in $\mathcal{T}_{M3}$. Here, Total Good = Good and Overlapping + Good and Exclusive.

It is worth noting here, that there are different taxonomy types in practice. *Grounded* taxonomies are typically revised over time as new empirical evidence is gathered through the scientific method. A recently accepted version in the literature can be considered closer to a ground truth than a conceptual taxonomy, in that it offers less scope for a language model to generate new terms absent new evidence. We use the following grounded taxonomies for evaluation: "*Animal Phylogenetic Classes*" (Bánki et al., 2025), "*Periodic Chemical Elements*" (International Union of Pure and Applied Chemistry (IUPAC), 2022), and "*Mineral Classification*" (Gaines et al., 1997-2024).

*Conceptual* taxonomies are more subjective than grounded ones; even the definitions and usage of terminology can vary across the literature (Usman et al., 2017; Szopinski et al., 2020; Kundisch et al., 2021; Kaplan et al., 2022). We use the following conceptual taxonomies for evaluation: "*Risk of Language Models*" (Weidinger et al., 2022), "*Online Harmful Content*" (Banko et al., 2020), and "*Logical Fallacies*" (Curtis, 2023)). Because of this subjectivity, we additionally compute the following metrics for conceptual taxonomies:

- **Novelty**: This metric assesses whether $\mathcal{T}_{M3}$ contains relevant nodes that are not present in $\mathcal{T}_E$. The LM critic identifies nodes in $\mathcal{T}_{M3}$ that are not semantically equivalent to any node in $\mathcal{T}_E$ and then

judges the relevance of these novel nodes. A higher number of relevant novel nodes indicates greater novelty. We define this as the ratio (Good and Exclusive in $\mathcal{T}_{\text{M3}}$) / (Total Good in $\mathcal{T}_E$).

- **Coverage**: This represents the total number of "good" items in $\mathcal{T}_{\text{M3}}$ relative to the total number of "good" items in $\mathcal{T}_E$. Coverage is equivalent to Completeness + Novelty and provides a comparative metric of the number of sound items. It follows that a coverage value greater than 1.0 indicates that $\mathcal{T}_{\text{M3}}$ covers more relevant and correct items than $\mathcal{T}_E$ within the global space for the given taxonomy.

A more elaborate description of the grounded and conceptual taxonomies used can be found in B.2.

### B.2 Taxonomy Evaluation Results

Table 3: **Performance metrics for different taxonomies.**

| Type | Topic | Method | Completeness | Soundness | Novelty | Coverage |
|------|-------|--------|--------------|-----------|---------|----------|
| Conceptual | Online Harmful Content | Simula | **0.749** | **0.980** | **0.865** | **1.614** |
| | | 0-shot | 0.588 | 0.957 | 0.412 | 1.000 |
| | Logical Fallacies | Simula | **0.726** | 0.919 | **1.679** | **2.405** |
| | | 0-shot | 0.458 | **0.966** | 0.193 | 0.651 |
| | Risks of Language Models | Simula | **0.867** | **1.000** | 0.267 | **1.134** |
| | | 0-shot | 0.467 | **1.000** | **0.367** | 0.834 |
| Grounded | Animal Phylogenetic Classes | Simula | **0.458** | **0.926** | — | |
| | | 0-shot | 0.349 | 0.918 | | |
| | Periodic Chemical Elements | Simula | **0.993** | **0.987** | — | |
| | | 0-shot | 0.775 | 0.864 | | |
| | Mineral Classification | Simula | **0.762** | **0.340** | — | |
| | | 0-shot | 0.442 | 0.329 | | |

Description of the taxonomies from Table 3:

- **[Conceptual] Online Harmful Content:** This typology aims to provide a unified classification of harmful content found online. It synthesizes common abuse types described by industry content policies, policy recommendations, community standards, and health expert guidelines. The goal is to create readily usable categories for content moderation, encourage the development of accurate datasets for model training, and raise awareness of less-studied abuse types to improve online safety. This taxonomy categorizes different types of harmful content found online into four main groups: Hate and Harassment; Self-Inflicted Harm; Ideological Harm; and Exploitation — and further branches each into a set of more specific types. (Banko et al., 2020).

- **[Conceptual] Logical Fallacies:** This taxonomy classifies types of logical fallacies into a hierarchical structure. It divides fallacies into two main branches: Formal Fallacies (errors in the structure of the argument) and Informal Fallacies (errors in the content or context of the argument). These main categories are further subdivided into numerous specific types of fallacies, such as Propositional Fallacies, Quantificational Fallacies, and various informal fallacies like Appeal to Ignorance, and Red Herring, and then branching down to increasingly specific types (Curtis, 2023).

- **[Conceptual] Risks of Language Models:** This taxonomy identifies ethical and social risks associated with large-scale language models (LMs). It categorizes these risks into six areas: Discrimination, Hate speech and Exclusion; Information Hazards; Misinformation Harms; Malicious Uses; Human-Computer Interaction Harms; and Environmental and Socioeconomic harms. The taxonomy distinguishes between "observed" risks (already evidenced in LMs) and "anticipated" risks

(considered likely but not yet observed). The goal is to provide a comprehensive framework for understanding and mitigating the potential negative consequences of LMs. (Weidinger et al., 2022).

- **[Grounded] Animal Phylogenetic Classes:** This taxonomy represents the hierarchical classification of animals based on their evolutionary relationships. It is truncated to two levels deeper into the animal kingdom, encompassing its phyla and classes. (Bánki et al., 2025).

- **[Grounded] Periodic Chemical Elements:** This taxonomy organizes chemical elements into a hierarchical structure based on their atomic number, electron configuration, and recurring chemical properties, primarily reflecting their placement in the periodic table. It positions elements into groups from Group 1 - Alkali Metals through Group 18 - Noble Gases, as well as the Lanthanides and Actinides. Each group is further lists individual elements like Sodium (Na) or Gold (Au). The structure represents the periodic trends and shared characteristics within groups, enabling chemists to understand relationships and predict elemental behavior. (International Union of Pure and Applied Chemistry (IUPAC), 2022).

- **[Grounded] Mineral Classification:** Presented in Dana's New Mineralogy, Eighth Edition, this taxonomy is a hierarchical classification system for minerals, employing a four-part numerical code to categorize each species. This system, analogous to the Linnaean taxonomy for biology, organizes minerals based on both their chemical composition and crystal structure. The first number denotes the mineral's class (e.g., anhydrous carbonates), reflecting broad compositional categories or dominant structural features (especially in silicates). The second number signifies the mineral's type, sometimes based on atomic properties, or formula. The third number groups minerals with similar structural arrangements. Finally, the fourth number uniquely identifies the individual mineral species, such as Calcite or Magnesite within the Calcite Group. This numerical system offers a structured and expandable framework, allowing new minerals to be easily integrated while highlighting the chemical and structural relationships between different mineral species. (Gaines et al., 1997-2024).

### B.3 Limitations

- **Preference Bias.** As discussed in the Related Work in Appendix A, there is evidence that M3s prefer model-generated text over text generated by humans. In our case, we do not prompt the M3 to express a preference. Rather, we ask if certain nodes semantically approximate other nodes, or if certain nodes are appropriate given the context. However, we did use models from the same model family for both the generation and the evaluation. Thus, future work could repeat this experiment with separate generator and critic models.

- **Stochastic Sensitivity.** We did not optimize prompts for each separate taxonomy. As such, the reported metrics likely represent lower bounds.

- **Downstream Application.** While we performed a comparative evaluation of synthetic and real taxonomies, it is not directly clear which are better suited for certain downstream applications.

### B.4 Taxonomy Generation Algorithm

---

**Algorithm 1** Reasoning-Driven Taxonomy Generation

---

**Require:** User instructions $y$, Target depth $D$, Multi-modal Model M3
**Ensure:** $\mathbb{T} = \{\mathcal{T}_i\}_{i=0}^{K}$, set representing conceptual space $\mathcal{Y}$

1: **Phase 1: Factor Disentanglement**
2:   $\triangleright$ *Extract prime factors of variation from instructions*
3:   $F \leftarrow$ M3.ProposeFactors$(y)$                                              $\triangleright F = \{f_0, \ldots, f_K\}$
4:   $\triangleright$ *Initialize set to store generated taxonomies*
5:   $\mathbb{T} \leftarrow \emptyset$

6: **Phase 2: Breadth-First Taxonomic Expansion**
7: **for each** factor $f_i \in F$ **do**
8:     $\triangleright$ *Initialize taxonomy $\mathcal{T}_i$ with root node*
9:     $n_{root} \leftarrow$ Node$(y, f_i)$
10:     $\mathcal{T}_i \leftarrow \{n_{root}\}$
11:     $\triangleright$ *Queue for nodes at current depth level*
12:     $Q_{curr} \leftarrow \{n_{root}\}$
13:     $\triangleright$ *Strategic plan guiding expansion consistency*
14:     $P \leftarrow$ "Expand based on $y$ and $f_i$"
15:     **for** depth $d = 1$ **to** $D$ **do**
16:        $\triangleright$ *Queue to collect nodes for the next depth*
17:        $Q_{next} \leftarrow \emptyset$
18:        **for each** node $n \in Q_{curr}$ **do**
19:           $\triangleright$ *Gather global and local reasoning context*
20:           $\mathcal{C} \leftarrow \{y, \text{Ancestors}(n), \text{Siblings}(n)\}$
21:           $\triangleright$ ***Step 1: Proposal (Best-of-$N$)***
22:           Query M3 $N$ times using context $\mathcal{C}$ and guidance Plan $P$
23:           $\mathcal{N}_{raw} \leftarrow \bigcup_{j=1}^{N}$ M3.Output$_j$                      $\triangleright$ *Set of raw child nodes*
24:           $\triangleright$ ***Step 2: Critic Refinement***
25:           $\triangleright$ *Prompt M3 to improve completeness, soundness, specificity*
26:           $\mathcal{N}_{refined} \leftarrow$ M3.Critique$(\mathcal{C}, \mathcal{N}_{raw})$          $\triangleright$ *Add, remove, merge, edit*
27:           Add $\mathcal{N}_{refined}$ as children of $n$ in $\mathcal{T}_i$
28:           $Q_{next} \leftarrow Q_{next} \cup \mathcal{N}_{refined}$
29:        $\triangleright$ ***Step 3: Planning (Per Level)***
30:        **if** $d < D$ **then**
31:           $\triangleright$ *Generate guidance for next level based on all new nodes*
32:           $P \leftarrow$ M3.GeneratePlan$(y, Q_{next})$
33:        $\triangleright$ *Advance to next depth*
34:        $Q_{curr} \leftarrow Q_{next}$
35:     $\mathbb{T} \leftarrow \mathbb{T} \cup \{\mathcal{T}_i\}$
36: **return** $\mathbb{T}$

---

## B.5 Qualitative Examples

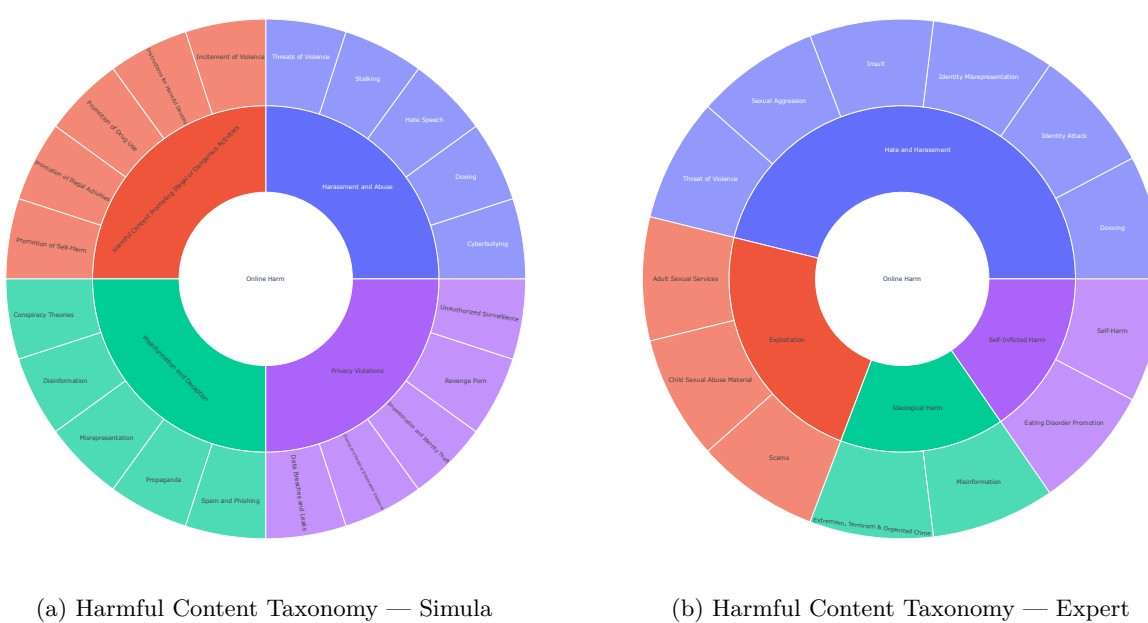

(a) Harmful Content Taxonomy — Simula

(b) Harmful Content Taxonomy — Expert

Figure 8: Comparison of Online Harmful Content Taxonomy (Simula vs. Expert).

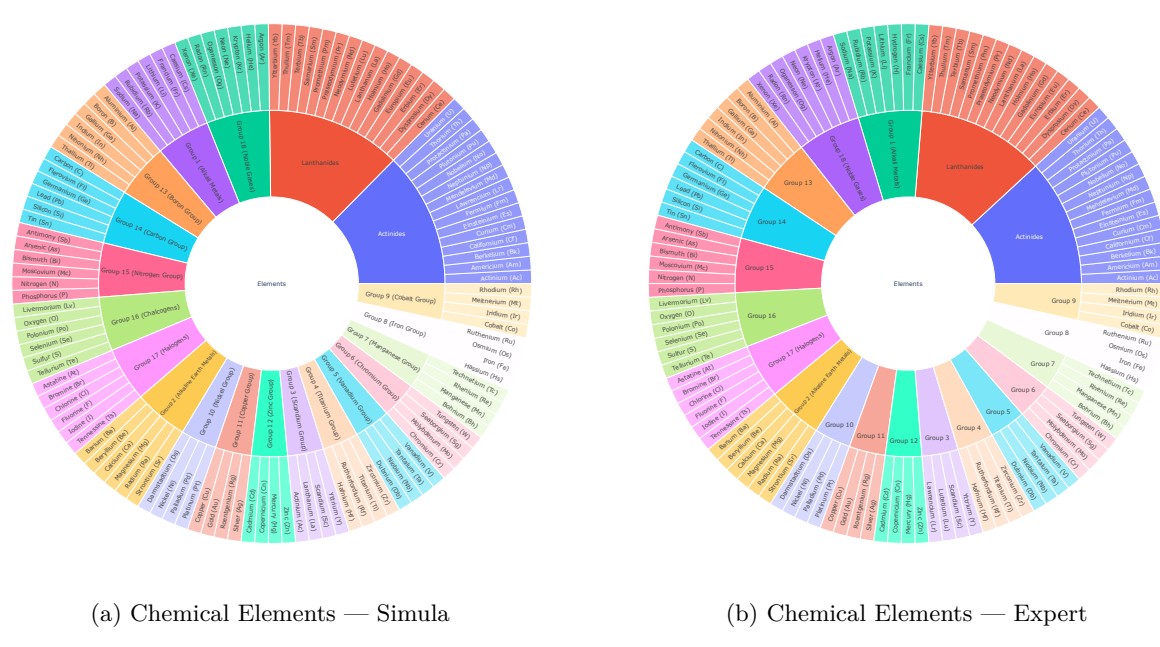

(a) Chemical Elements — Simula

(b) Chemical Elements — Expert

Figure 9: Comparison of Chemical Element Taxonomy (Simula vs. Expert).

## C  Synthetic Data Generation Algorithm

---

**Algorithm 2** Agentic Data Generation

---

**Require:** Taxonomies $\mathbb{T}$, User instructions $y$, Target size $N$, Model M3, Complexity ratio $c \in [0, 1]$, Sampling Strategies $\mathcal{S}$, Scenarios per mix $K$
**Ensure:** Synthetic Dataset $\mathcal{D}$ containing exactly $N$ high-quality data points

1: **Initialization**
2:   ▷ *Initialize set to store validated data points*
3:   $\mathcal{D} \leftarrow \emptyset$

4: **Generation Loop**
5:   ▷ *Continue until target size is reached*
6: **while** $|\mathcal{D}| < N$ **do**
7:     ▷ ***Step 1: Global Diversity Sampling***
8:       ▷ *Sample a single "Mix" (combination of nodes) based on strategies*
9:     $\mathcal{M} \leftarrow \text{SampleSingleMix}(\mathbb{T}, \mathcal{S})$
10:    ▷ ***Step 2: Meta-Prompt Generation (Local Diversity)***
11:      ▷ *Generate K diverse scenarios (meta-prompts) based on the Mix*
12:    $\mathcal{P}_{scenarios} \leftarrow \text{M3.GenerateScenarios}(y, \mathcal{M}, K)$
13:      ▷ *Choose one scenario at random*
14:    $p_{meta} \leftarrow \text{RandomSample}(\mathcal{P}_{scenarios})$
15:    ▷ ***Step 3: Complexification***
16:    **if** $\text{Random}(0, 1) < c$ **then**
17:        ▷ *Inject constraints or edge-cases to increase difficulty*
18:        $p_{meta} \leftarrow \text{M3.Complexify}(p_{meta})$
19:    ▷ ***Step 4: Generation and Critic-Refinement Loop***
20:      ▷ *Generate initial proposal based on meta-prompt*
21:    $d_{point} \leftarrow \text{M3.Generate}(p_{meta})$
22:    $is\_satisfying \leftarrow$ **false**
23:    **repeat**
24:        ▷ *Critic evaluates if data point fulfills meta-prompt requirements*
25:        $(verdict, explanation) \leftarrow \text{M3.Critique}(p_{meta}, d_{point})$
26:        **if** $verdict ==$ "satisfying" **then**
27:            $is\_satisfying \leftarrow$ **true**
28:        **else**
29:            ▷ *Agentic refinement based on critic's explanation*
30:            $d_{point} \leftarrow \text{M3.Refine}(p_{meta}, d_{point}, explanation)$
31:    **until** $is\_satisfying$ **or** max retries reached
32:    **if** $is\_satisfying$ **then**
33:        ▷ *Only add to dataset if it passed the critic*
34:        $\mathcal{D} \leftarrow \mathcal{D} \cup \{d_{point}\}$
35: **return** $\mathcal{D}$

---

# D  Evaluating Double-Critic Steps

## D.1  Multi-Lingual MMLU

In Table 4 we show empirical critic-rejection sampling results for a subset of MMLU questions on Mathematics, Computer Science, and Physics. We use the subjects' education levels (elementary, high-school, and college) as ground truth complexity categories. We evaluate performance on languages with different resource categories, e.g., "Low", "Mid", and "High", according to their recorded, written, and cataloged NLP resources (Singh et al., 2024b). We observe that our critic-rejection sampling strategy is effective for each language under each complexity condition.

Table 4: **Critic Rejection Sampling on Multi-Lingual MMLU Questions.** We evaluate our critic-rejection sampling method for MMLU questions on Mathematics, Physics, and Computer Science. We use the subject education level (elementary, high-school, and college) as the ground-truth Complexity. We display the realized change in accuracy, $\mu_{\text{gen}}$ of following critic rejections. Also shown are the average Elo complexity score and the size of the rejected and accepted subsets.

| Complexity | Critic | English (High) | | | Korean (Mid) | | | Nepali (Low) | | |
|---|---|---|---|---|---|---|---|---|---|---|
| | | $\mu_{\text{gen}}$ | Elo | $|\mathcal{D}|$ | $\mu_{\text{gen}}$ | Elo | $|\mathcal{D}|$ | $\mu_{\text{gen}}$ | Elo | $|\mathcal{D}|$ |
| Level 1 | $\times$ | $0.93 \ _{\pm 0.07}$ | $290 \ _{\pm 8}$ | 14 | $0.86 \ _{\pm 0.07}$ | $306 \ _{\pm 7}$ | 29 | $0.76 \ _{\pm 0.07}$ | $308 \ _{\pm 7}$ | 41 |
| | $\checkmark$ | $0.98 \ _{\pm 0.01}$ | $303 \ _{\pm 2}$ | 364 | $0.97 \ _{\pm 0.01}$ | $301 \ _{\pm 2}$ | 349 | $0.97 \ _{\pm 0.01}$ | $304 \ _{\pm 2}$ | 337 |
| Level 2 | $\times$ | $0.52 \ _{\pm 0.07}$ | $427 \ _{\pm 7}$ | 46 | $0.64 \ _{\pm 0.06}$ | $428 \ _{\pm 5}$ | 53 | $0.58 \ _{\pm 0.06}$ | $431 \ _{\pm 6}$ | 60 |
| | $\checkmark$ | $0.94 \ _{\pm 0.01}$ | $427 \ _{\pm 2}$ | 578 | $0.92 \ _{\pm 0.01}$ | $426 \ _{\pm 2}$ | 571 | $0.93 \ _{\pm 0.01}$ | $425 \ _{\pm 2}$ | 564 |
| Level 3 | $\times$ | $0.67 \ _{\pm 0.10}$ | $473 \ _{\pm 5}$ | 24 | $0.54 \ _{\pm 0.10}$ | $471 \ _{\pm 5}$ | 26 | $0.41 \ _{\pm 0.09}$ | $473 \ _{\pm 4}$ | 34 |
| | $\checkmark$ | $0.89 \ _{\pm 0.02}$ | $467 \ _{\pm 2}$ | 278 | $0.88 \ _{\pm 0.02}$ | $468 \ _{\pm 2}$ | 276 | $0.86 \ _{\pm 0.02}$ | $465 \ _{\pm 2}$ | 268 |

# E    Calibrated Complexity Scoring

We compare model-assigned complexity scores against the ground-truth human annotations. We ablate model-assigned complexity scores varying the number of times each sample is scored (**N**) and the batch size (**BS**) of questions being scored simultaneously. Importantly, for fixed N, the number of samples being scored simultaneously (BS) increases the context length but reduces the number of inference passes. For example, for $|\mathcal{D}| = 1000$, N=10 and BS=1, we require $10,000$ inference passes. Setting BS to 5 instead reduces this to $10,000/5 = 2,000$. All things equal, in practice we would thus like to see a higher BS to have similar or better performance than a lower BS.

## E.1    Open Generation: MATH

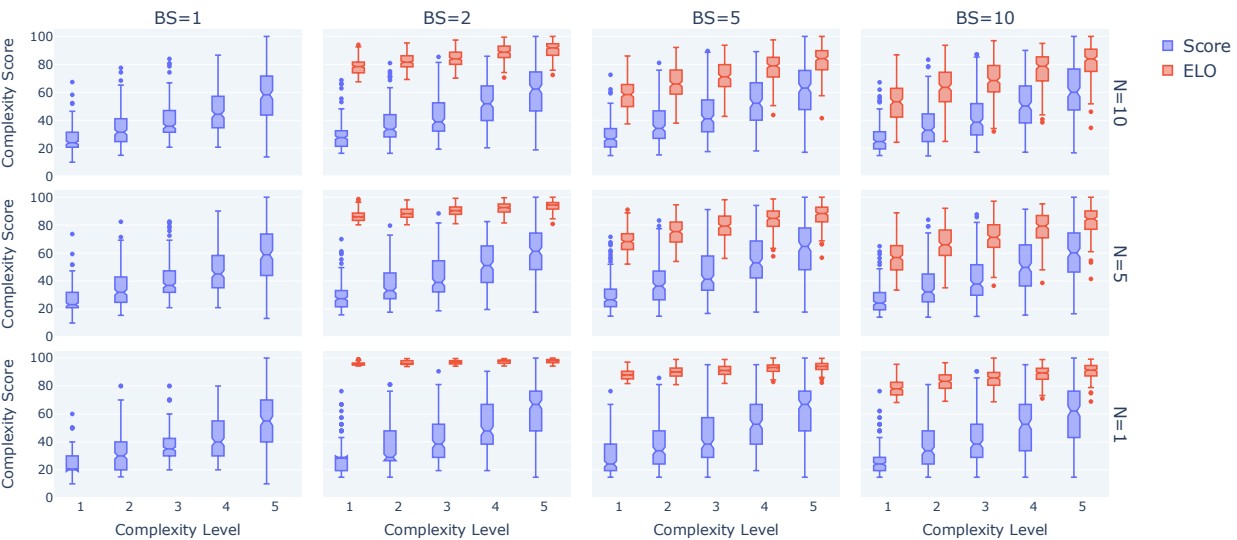

Figure 10: **Complexity score ablation MATH test set.** The MATH dataset comes with ground truth complexity levels ranging from 1-5. We group by the ground truth complexity level and plot the model-assigned complexity scores for each group. We compare using average raw scores (Score) against using Elo rankings (Elo) and ablate the batch size (**BS**) of question scored simultaneously and the number of times each question is scored (**N**). All results are scaled per (BS, N)-grouping to lie between 0 and 100. For $BS = 1$, no pairwise rankings are done, so Elo rankings do not apply.

In Figure 10, we show results of comparing model-assigned complexity scores to the human-annotated ground truth (Levels 1-5). To enable side-by-side comparison, we scale both the raw average scores (Score) and the computed ELO rankings (ELO) to lie between 0 and 100 for each {BS, N} grouping. As we increase the number of samples, N, clusters become better separated. Increasing BS > 1 enables the use of latent skill methods like ELO to increase consistency. We found BS = N = 5 to strike an appropriate balance between separation and inference cost.

## E.2    Multiple-Choice Generations: MMLU Global

In Table 5 we compare model-assigned complexity scores for a subset of MMLU questions on Mathematics, Computer Science, and Physics. We use the subjects' education levels (elementary, high-school, and college) as ground truth complexity categories. We evaluate performance on languages with different resource categories, e.g., "Low", "Mid", and "High", according to their recorded, written, and catalogued NLP resources per Singh et al. (2024b). After running KMeans on the model-assigned complexity scores, we compute the Normalized

Mutual Information (**NMI**) and the Adjusted Rand Index (**ARI**) to evaluate cluster approximation of the ground truth complexity. Finally, we train a logistic regression on model-assigned complexity scores to evaluate them as an estimator for the model's generative performance, reporting the Area Under the Curve (**AUC**). Similar to our findings in Appendix D.1, we find model-assigned complexity scoring robust across several languages. Choosing BS = N = 5 again emerge as reasonable hyper-parameters.

Table 5: **Multi-lingual MMLU Complexity Scoring.** We use exam questions for the topics Mathematics, Physics, and Computer Science, on education levels elementary (Mathematics only), high-school, and college. Taking the education level as our ground-truth complexity level (1-3), we run KMeans on the complexity scores generated by the model and compute the Normalized Mutual Information (**NMI**) and the Adjusted Rand Index (**ARI**). Finally, we compute the Area Under the Curve (**AUC**) of using the complexity scores as an estimator for the model's generative performance. We ablate the batch size (**BS**) of exam questions scored simultaneously and the number of times each exam question is scored (**N**).

| BS | N | English (High) | | | Arabic (High) | | | Dutch (High) | | | Korean (Mid) | | | Nepali (Low) | | |
|---|---|---|---|---|---|---|---|---|---|---|---|---|---|---|---|---|
| | | NMI | ARI | AUC | NMI | ARI | AUC | NMI | ARI | AUC | NMI | ARI | AUC | NMI | ARI | AUC |
| 1 | 1 | 0.27 | 0.23 | 0.61 | 0.32 | 0.27 | 0.60 | 0.32 | 0.28 | 0.59 | 0.32 | 0.27 | 0.59 | 0.32 | 0.27 | 0.56 |
| | 5 | 0.27 | 0.24 | 0.62 | 0.33 | 0.28 | 0.61 | 0.34 | 0.30 | 0.59 | 0.34 | 0.29 | 0.59 | 0.33 | 0.29 | 0.56 |
| 5 | 1 | 0.36 | 0.31 | 0.63 | 0.38 | 0.32 | 0.60 | 0.37 | 0.31 | 0.59 | 0.38 | 0.33 | 0.59 | 0.37 | 0.32 | 0.57 |
| | 5 | 0.42 | 0.36 | 0.64 | 0.42 | 0.35 | 0.62 | 0.41 | 0.35 | 0.59 | 0.40 | 0.34 | 0.60 | 0.41 | 0.35 | 0.57 |
| 10 | 1 | 0.38 | 0.33 | 0.64 | 0.39 | 0.34 | 0.61 | 0.37 | 0.32 | 0.60 | 0.40 | 0.35 | 0.61 | 0.39 | 0.33 | 0.57 |
| | 5 | 0.42 | 0.37 | 0.63 | 0.41 | 0.36 | 0.62 | 0.40 | 0.34 | 0.60 | 0.43 | 0.38 | 0.61 | 0.43 | 0.37 | 0.57 |

### E.3 Calibrated Complexity Scoring Algorithm

---

**Algorithm 3** Calibrated Attribute Scoring

---

**Require:** Dataset $\mathcal{D}$, Attribute guidance $\mathcal{A}$, Model M3, Samples per item $K$, Batch size $BS$
**Ensure:** Scoring mapping $\mathcal{S}_{final} : d \rightarrow (\text{score}_{raw}, \text{rank}_{elo})$ for all $d \in \mathcal{D}$

1:  **Phase 1: Batch Preparation**
2:  ▷ *Create batch schedule ensuring every item appears in $\approx K$ varied batches*
3:  $\mathbf{B} \leftarrow \text{PrepareBatches}(\mathcal{D}, BS, K)$
4:  ▷ *Initialize map to store lists of raw scores for each item*
5:  $RawScores \leftarrow \{d : [] \text{ for } d \in \mathcal{D}\}$

6:  **Phase 2: Batch-wise Relative Scoring**
7:  **for each** batch $B \in \mathbf{B}$ **do**
8:  ▷ *Prompt M3 to score items relative to others within this batch*
9:      $Scores_B \leftarrow \text{M3.ScoreSingleBatch}(B, \mathcal{A})$
10: ▷ *Collect raw scores for aggregation*
11:     **for each** $(d, score) \in Scores_B$ **do**
12:         Append $score$ to $RawScores[d]$

13: **Phase 3: Score Calibration (Elo Reranking)**
14: ▷ *Derive pairwise comparisons from batches to compute global Elo ratings*
15: $EloRatings \leftarrow \text{CalibrateScores}(RawScores, \mathbf{B})$

16: **Phase 4: Aggregation and Finalization**
17: $\mathcal{S}_{final} \leftarrow \emptyset$
18: **for each** $d \in \mathcal{D}$ **do**
19: ▷ *Compute average raw score across $K$ appearances*
20:     $\mu_{raw} \leftarrow \text{Mean}(RawScores[d])$
21: ▷ *Retrieve globally calibrated rank*
22:     $r_{elo} \leftarrow EloRatings[d]$
23:     $\mathcal{S}_{final}[d] \leftarrow (\mu_{raw}, r_{elo})$
24: **return** $\mathcal{S}_{final}$

---

# F    Downstream Experiments

## F.1    Training Configuration and Hyperparameters

We first conducted a hyperparameter sweep to identify the optimal LoRA rank and batch size for each dataset. Checkpoint selection was based on the validation accuracy of the final answer, where we chose the checkpoint corresponding to the peak of a 3-point moving average of scores over the final evaluation steps. This process yielded a fixed set of hyperparameters for each dataset. However, since the optimal learning rate proved sensitive to the training data size, we retained a sweep over two values (0.005, 0.01) for our final experiments. These final runs were performed with 10 different seeds for each configuration. For each data size, we then selected the runs from the best-performing learning rate on average for that specific system version.

We use the Adafactor optimizer for training Shazeer & Stern (2018). To ensure stability, we apply global gradient clipping with a maximum norm of 1.0. The learning rate follows a linear warm-up, followed by cosine decay schedule. The number of warm-up steps is calculated dynamically based on the total training steps (still bounded between a minimum of 500 and a maximum of 5,000 steps)

Table 6: Final hyperparameters used for each dataset. The final learning rate was chosen from the specified set based on validation performance for each training data size.

| Dataset | Learning Rates | LoRA Rank | Batch Size |
|---------|----------------|-----------|------------|
| CTI RCM | {0.005, 0.01} | 8 | 16 |
| CTI MCQ | {0.005, 0.01} | 8 | 16 |
| GSM8K | {0.005, 0.01} | 16 | 16 |
| LEXam | {0.005, 0.01} | 16 | 16 |
| MMLU | {0.005, 0.01} | 16 | 16 |

## F.2    Additional Evaluation Results

In Figure 11, we show the results on two additional subsets of Global MMLU (Physics, Math, and Computer Science). The trend shown in Section 4.3 for the Nepali subset continues to be the case here, where the full Simula system shows a better accuracy growth trend with larger datasets.

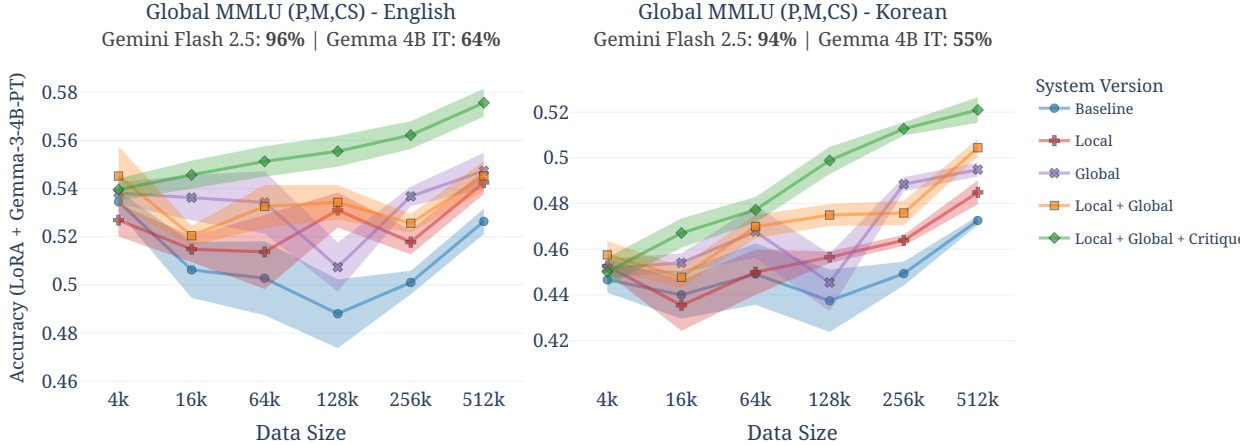

Figure 11: **Downstream Performance on Global MMLU English and Korean.** We report mean accuracy with 95% CI for these additional datasets. We continue to see that the full Simula system version with the critic, provides the best performance compared to the other versions.

### F.3 Choice of Optimizer

As detailed in the Section F.1, we use Adafactor as our optimizer for all downstream fine-tuning experiments. A reasonable concern would be if the choice of optimizer could influence the reported results in Section 4.3. To address these we conducted a targeted ablation and offer additional theoretical justification below.

We replicated the downstream experiment on the CTI MCQ dataset, scaling the data size from 4k to 84k using the Muon optimizer (Jordan et al., 2024). We opted for the adaptive Muon optimizer because of its structural distinctness from Adafactor (i.e., momentum-orthogonal vs. factored second-moment). As shown in Figure 12, we observe that the relative performance trends hold: The Full Simula system consistently outperforms the Baseline and different variants across data sizes. These results suggest that the benefits provided by our reasoning-driven approach are robust to the choice of optimizer.

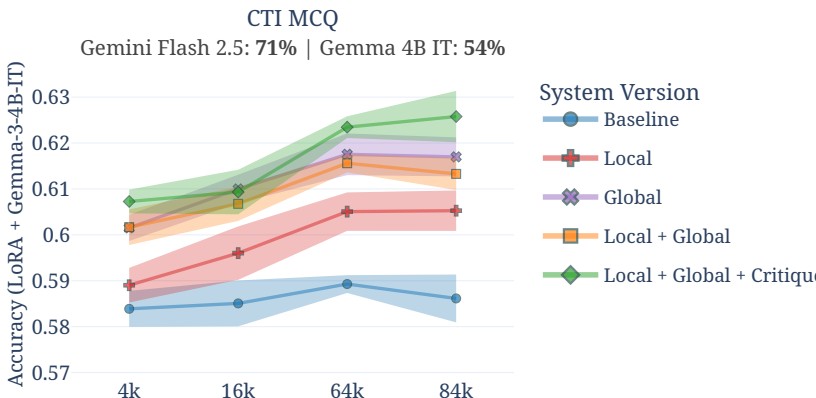

Figure 12: **Downstream Performance using Muon Optimizer.** We report mean accuracy with 95% CI for the CTI MCQ dataset for student models fine-tuned using the Muon optimizer. We continue to see that the full Simula system version with the critic, provides the best performance compared to the other versions.

**Informed Hyperparameter Selection.** The reported results in Section 4.3 using Adafactor are not based on default settings, but the outcome of rigorous hyperparameter sweeps over data sizes and configurations (see Appendix F.1). As noted in (Schmidt et al., 2021), while optimizer performance can vary greatly across tasks, the authors show that evaluating multiple optimizers with default parameters works approximately as well as tuning the hyperparameters of a single, fixed optimizer.

**Scaling Laws for Relative Trends.** The goal of our ablations is to measure the *relative* performance differences of the different system components. We can therefore distinguish between the *exponent* of the scaling law (rate at which error decreases with data) and the *intercept* (training efficiency). The former is shown to be strongly correlated with data composition (Sorscher et al., 2022; Chen et al., 2025), while the latter is influenced by optimization and architecture (Schmidt et al., 2021; Everett et al., 2024). Since our claims focus on optimal scaling trajectories (exponent) of Simula data, the choice of optimizer (primarily intercept) should not invalidate the relative advantages observed.

### F.4 Choice of the Teacher Model

Throughout this work we used Gemini 2.5 Flash as the generator model. To demonstrate that Simula is a model-agnostic framework we conducted a targeted ablation using the open-source *Qwen3-Next-80B Instruct* model (Yang et al., 2025). We replicated the downstream experiment on the CTI MCQ dataset, scaling the generated data size from 4k to 84k, while keeping the student model and optimization parameters unchanged.

As shown in Figure 13 below, performance trends remain consistent with the results reported in Section 4.3 using the Gemini model: student models fine-tuned on data generated by the Full Simula system consistently outperform the Baseline and partial ablations across data scales. This provides additional evidence that the observed improvements are due to our reasoning-driven data generation approach, rather than model-specific factors.

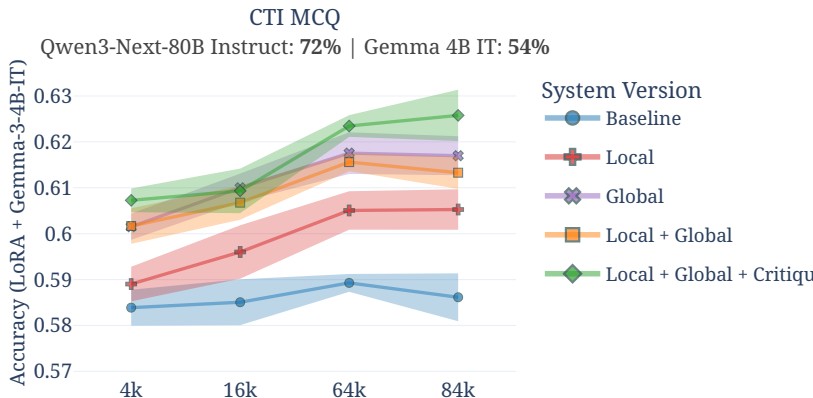

Figure 13: **Downstream Performance using Qwen3-Next-80B Instruct.** We report mean accuracy with 95% CI for the CTI MCQ dataset for data generated using the Qwen-3 80B instruction-tuned model. We continue to see that the full Simula system version with the critic, provides the best performance compared to the other versions.

