# OpenReview forum: "Reasoning-Driven Synthetic Data Generation and Evaluation"
_TMLR — Accepted by TMLR_

### Review · Reviewer_T615 · 2025-11-16

**Summary Of Contributions:**

This paper introduces Simula, a reasoning-first, seedless, and agentic framework for generating high-quality synthetic datasets at scale. The authors argue that existing synthetic data approaches struggle with three major issues: (1) unclear definitions of “good” data (quality, diversity, complexity), (2) lack of transparent and controllable generation mechanisms, and (3) limited or misleading evaluation practices relying heavily on noisy or incomplete benchmarks. Simula is designed to address these gaps by building each synthetic example through explicit reasoning and structured conceptual decomposition.

**Audience:**

Yes

**Audience Explanation:**

At least a portion of TMLR’s audience would be interested in this paper’s findings. The work addresses an increasingly important problem in modern ML: how to generate high-quality synthetic data in a controllable and scalable way. This topic is relevant to researchers working on data-centric model training, evaluation methodology, LLM alignment, and multimodal system development. The paper’s insights on diversity, complexity, and mechanism design also speak to broader issues in dataset construction and benchmarking, which are central concerns for TMLR readers. Even if the framework is specialized, the empirical findings and methodological contributions would likely appeal to those studying data quality, model scaling, and synthetic data pipelines.

**Claims And Evidence:**

Yes

**Claims Explanation:**

This paper presents a well-designed and thorough investigation into reasoning-driven synthetic data generation. The proposed Simula framework is technically novel in its focus on transparency, controllability, and seedless generation, offering a clear mechanism design that is rarely discussed in synthetic-data research. The paper introduces several strong methodological ideas, including taxonomy-based coverage, meta-prompt construction, and a double-critic refinement process, which together form a coherent and interpretable pipeline. The experimental section is extensive and carefully executed: the authors evaluate on both niche and widely used benchmarks, conduct detailed ablations of each system component, study scaling behavior, and propose new intrinsic metrics such as taxonomy coverage and calibrated complexity scoring. These analyses provide valuable insights into how diversity, complexity, and correctness influence downstream model performance.

**Requested Changes:**

I would like to see more discussion or evidence regarding whether the reported downstream trends depend on the choice of optimizer. The current results are presented using Adafactor, but generalization behavior can vary meaningfully across optimizers, especially in low-data or synthetic-data regimes. Prior work has shown that optimizers such as AdamW, Muon, and other adaptive variants can yield different stability and generalization characteristics even when trained on identical data. It is therefore unclear whether the scaling patterns and conclusions drawn in this paper would remain consistent under different optimization settings. I encourage the authors to either provide additional experiments with alternative optimizers or include a more explicit justification for why the observed effects should be robust to optimizer choice.

---

> ### Author Response · Authors · 2025-12-10
>
> We would like to thank Reviewer T615 for the positive review and appreciate the suggestion to investigate optimizer robustness in our downstream experiments. To address this while avoiding the prohibitive costs of repeating all experiments, we conducted a targeted ablation and offer additional theoretical justification below. We uploaded new supplementary material (Supplementary Material - Reviewer T615) containing the requested empirical ablation and the theoretical discussion.
>
> **New Empirical Evidence**: We replicated the downstream experiment on the CTI MCQ dataset, scaling the data size from 4k to 84k using the Muon optimizer [1]. We opted for the Muon optimizer suggested by the reviewer because of its structural distinctness from Adafactor (momentum-orthogonal vs. factored second-moment). As shown in the uploaded supplementary material, we observe that the relative performance trends hold: The Full Simula system consistently outperforms the Baseline and different variants across data sizes. These results suggest that the benefits provided by our reasoning-driven approach are robust to the choice of optimizer.
>
> **Additional Justification**:
> - **Informed Hyperparameter Selection**: The reported results in Section 4.3 using Adafactor are not based on default settings, but the outcome of rigorous hyperparameter sweeps over data sizes and configurations (see Appendix F.1). As noted in [2], while optimizer performance can vary greatly across tasks, the authors show that evaluating multiple optimizers with default parameters works approximately as well as tuning the hyperparameters of a single, fixed optimizer.
> - **Scaling Laws for Relative Trends**: The goal of our ablations is to measure the _relative_ performance differences of the different system components. We can therefore distinguish between the _exponent_ of the scaling law (rate at which error decreases with data) and the _intercept_ (training efficiency). The former is shown to be strongly correlated with data composition [3, 4], while the latter is influenced by optimization and architecture [2, 5]. Since our claims focus on optimal scaling trajectories (exponent) of Simula data, the choice of optimizer (primarily intercept) should not invalidate the relative advantages observed.
>
>
> We will update the manuscript to include the new empirical results and discussion regarding optimizer considerations.
>
>
>
> **References**:
>
> [1] Jordan et al., Muon: an optimizer for hidden layers in neural network, 2024
>
> [2] Schmidt et al., Descending through a crowded valley – benchmarking deep learning optimizers, ICML 2021.
>
> [3] Sorscher et al., Beyond neural scaling laws: beating power law scaling via data pruning, NeurIPS, 2022
>
> [4] Chen et al., Revisiting scaling laws for language models: the role of data quality and training strategies, ACL, 2025
>
> [5] Everett et al., Scaling exponents across parameterizations and optimizers, ICML, 2024

---

### Review · Reviewer_tfQe · 2025-11-25

**Summary Of Contributions:**

The paper introduces Simula, a framework for generating synthetic data that prioritizes explainability, control, and scalability over traditional stochastic or seed-based methods. The core methodology involves a multi-stage pipeline: (1) generating taxonomies to map the conceptual space of a domain, which ensures global diversity, (2) using agentic workflows to generate meta-prompts and complexify them, and (3) employing a dual-critic system to filter for quality and correctness.

The authors evaluate this framework intrinsically (via embedding diversity, taxonomic coverage, and a novel calibrated complexity scoring via Elo) and extrinsically (via downstream fine-tuning performance). Experiments are conducted across five datasets ranging from popular benchmarks (GSM8k, MMLU) to niche domains (Cyber Threat Intelligence, Legal exams).

Strength:
- The shift from random sampling to a taxonomy-driven approach addresses the common issue of mode collapse in synthetic data generation effectively.
- The discovery that increased data complexity does not always yield better downstream results (specifically when the teacher-student gap is small or the teacher is weak, as seen in LEXam) is valuable.
- The proposal of Taxonomic Coverage and Calibrated Complexity Elo offers concrete alternatives to vague embedding-similarity metrics.



Weaknesses:

- The experiments rely exclusively on Gemini 2.5 Flash for the teacher model and Gemma 3 4B for the student model.
- The primary baseline is a simplified version of their own pipeline (direct sampling). A comparison against established synthetic data methods is missing.

**Audience:**

Yes

**Audience Explanation:**

The work focuses on synthetic data generation, which is a core interest of the TMLR audience.

**Broader Impact Concerns:**

I have no broader impact concerns.

**Claims And Evidence:**

Yes

**Claims Explanation:**

The authors provide a rigorous experimental setup to support their claims. The ablation studies in Figure 6 show that combining Local + Global + Critique consistently outperforms the Baseline and partial implementations across data scales.

**Requested Changes:**

- Generalization Across Model Families: The paper relies entirely on Gemini/Gemma. To demonstrate that Simula is a generalizable framework and not specific to Gemini, it is important to present an ablation study using a different model family (e.g., Llama-3) for the generator/teacher role.

- Comparison to Baselines: The current baseline is essentially a stripped-down version of Simula. Please include the comparison against standard synthetic data generation methods.

---

> ### Author Response · Authors · 2025-12-10
>
> We thank Reviewer tfQe for the thoughtful feedback and acknowledging Simula’s contributions. We address the two requested changes below and uploaded new supplementary material containing the requested empirical evidence (Supplementary Material - Reviewer tfQe):
>
> **Generalization across Model Families**: The reviewer correctly notes that our work relies on the Gemini family as the teacher model. To demonstrate that the Simula approach is agnostic to the underlying model while being mindful of computational costs, we conducted a targeted ablation using the open-source Qwen-3 80B instruct model:
> - We replicated the downstream experiment on the CTI MCQ dataset, scaling the generated data size from 4k to 84k, while keeping the student model and optimization parameters unchanged.
> - As shown in the new supplementary material, performance trends remain consistent with our reported results using Gemini 2.5 flash as the teacher: students fine-tuned on the full Simula system consistently outperform the baseline and partial ablations across data scales.
> - This provides additional evidence that improvements are driven by our reasoning-driven data generation approach, rather than model-specific factors.
>
> **Interpreting our Baseline and Variants**: The reviewer requests a “_comparison against standard synthetic data generation methods_”. We would like to clarify that our baseline and ablation components are designed to serve as controlled proxies that have parallels to some popular “standard” approaches, allowing a fairer “apples-to-apples” comparison.
> - **Baseline**: Our baseline, which samples top-level nodes from a taxonomy and generates data directly, simulates a standard zero-shot/few-shot generation approach. We used top-level node sampling instead of sampling data directly using just a dataset description to avoid the severe mode collapse typically observed at scale.
> - **Local**: The local variant, which uses meta-prompting and complexification, conceptually maps to evolutionary or self-instruct approaches (e.g., [1, 2, 3]) that rely on iterative rewriting to increase diversity and complexity.
>
> A challenge in comparing Simula to “standard” techniques, is that these generally rely heavily on seed data or elaborate manual prompting. To the best of our knowledge, there were no standard, seedless baselines for seedless generation at scale at the time of project development. In their absence, we thus opted for careful ablations of the various system components to isolate specific mechanisms (e.g., taxonomic planning vs. random sampling, or meta-prompting vs. direct generation) and avoid the potential confounding effects of prompt engineering decisions.
>
> We will update our manuscript to include the Qwen 3 ablation and make the connections between the different system ablations clearer.
>
> **References**:
>
> [1] Wang, Yizhong, et al. Self-instruct: Aligning language models with self-generated instructions. ACL, 2023.
>
> [2] Xu, Can, et al. WizardLM: Empowering large pre-trained language models to follow complex instructions. ICLR, 2024.
>
> [3] Gupta, Himanshu, et al. Targen: Targeted data generation with large language models. COLM, 2024.

---

### Review · Reviewer_j5DE · 2025-12-02

**Summary Of Contributions:**

The authors lay out the main challenges of large-scale synthetic data generation, and then propose their framework Simula for addressing them.

The approach works as follows: a multi-modal model (M3) derives "taxonomies", which in my limited experience in the field believe to be semantic hierarchies of labels describing a particular qualitative factor of dataset variation. Once the taxonomies are derived, the framework can sample from them intelligently, such as by using a planning algorithm, thereby varying the different factors driving data generation in a controllable way (as opposed to uncontrolled generation which may have biases or uneven coverage of certain concepts). The data generation itself is augmented by a critic model to improve the output quality.

The authors benchmark their approach in various language-modelling settings, and show that each of their proposed changes (taxonomy sampling methods, use of critic, etc.) has a real benefit, as their experiments are thorough and include comparisons to a simple baseline.

My main concern with the work is that the concept of a "taxonomy", which is crucial to understanding the working mechanisms of Simula, is not clearly explained. In Section 2.1, the article offers descriptive examples of what "factors" are (ie. “cat type”), but does not go into details about how these factors are broken down into taxonomy objects. It may be useful to also offer an example of what the taxonomies related to these factors are, as only the appendix contains examples. Likewise, Figure 1 is hard to interpret without a-priori knowledge of what it is trying to describe and could benefit from annotations of "clusters" and visual explanation of how breaking into strict taxonomies creates the grid-like structure that covers the whole space more evenly ((b) and (c) are lacking the data points from (a) so it is hard to intuit what happens to them). More broadly, the paper does not define the term "taxonomy" prior to its use. It is my understanding that a taxonomy is a tree of concrete class labels/descriptions related to the corresponding factor.

**Audience:**

Yes

**Audience Explanation:**

Yes, synthetic data generation is a useful area in machine learning, and as the authors point out, highly relevant to modalities other than just language as well.

**Broader Impact Concerns:**

The ability to more strongly control data generation has obvious broader impact concerns. For example, it may facilitate the generation of nefarious content, or reinforce potential biases of the agent (human or M3) steering the generation process. It would be worth drawing attention to these concerns in a broader impact statement.

**Claims And Evidence:**

Yes

**Claims Explanation:**

Yes, the results are backed by significant number of benchmarks which are accurately represented by the paper's main claims. There is a strong focus on ablation study/comparison to the baseline so it is clear that the proposed method causes the claimed improvements in performance.

**Requested Changes:**

More clarity on the definition of "taxonomy" I believe is necessary for my recommendation. It is a key component of the Simula framework so readers, especially those with little background in the area, can understand the main problems it tries to solve. Currently, the term "taxonomy" is used before it is introduced/defined to the reader.

---

> ### Author Response · Authors · 2025-12-10
>
> We would like to thank Reviewer j5DE for their positive assessment of our experimental rigor and for identifying the potential confusion surrounding the concept of “taxonomy.” We agree that clarifying the definition would improve accessibility. We address the requested changes and broader impact concerns below and have attached new supplementary material (Supplementary Material - Reviewer j5DE) including: (i) a revision to section 2.1 and (ii) a new Broader Impact Statement.
>
> **On Taxonomies:**
> We agree that the term “taxonomy” is central to Simula’s approach and that the paper would benefit from additional background and examples to make it more accessible to a broader audience. To address this, we revised Section 2.1 to explicitly define a taxonomy as a "_hierarchical tree structure where the root node represents a broad factor of variation, and child nodes represent increasingly specific sub-categories or instantiations._"
>
> - **Expand definition**: We inserted a detailed definition immediately following Equation 1. This text explains that a taxonomy serves as a structured map of the concept space, breaking down abstract factors into concrete, sample-able attributes.
> - **Concrete Example**: As requested, we integrated the “cat type” example directly into the text to contextualize the algorithm described in Section 2.1 (see also Appendix B.4). For instance, we illustrate a taxonomy branch path as: “_cat type → domestic → shorthair → British shorthair_”. This will make it easier for the reader to visualize how abstract factors like “cat type” are broken down into granular taxonomy objects.
> - **Clarify Figure 1**: We updated the caption of Figure 1 to explicitly link the grid-like structure in b-c to the taxonomy structures. We clarified that the “grid” represents the discrete semantic space defined by a taxonomy’s leaf nodes at increasing granularity. For instance, the outer square of (b) might represent “cat type”, whereas a much smaller square would represent a specific breed like the “British shorthair.” This clarifies how using taxonomies allows for planned, even coverage, in contrast to the random clusters seen in (a).
>
> Finally, we can provide more detailed taxonomies from experiments as additional supplementary materials upon request. Unfortunately, the paper is not an optimal medium to visualize several levels of descriptive breakdowns.
>
>
> **On Broader Impact:**
> We appreciate the reviewer raising the concern that the steerability provided by Simula could be misused. Simula is a general-purpose method for generating synthetic data. While our work is motivated by a desire to increase control and steerability to overcome latent biases, we acknowledge that bad actors could use these tools to reinforce specific biases instead. We added a “Broader Impact Statement” to the paper acknowledging this dual-use risk while emphasizing the mitigation tools presented in our work:
> - **Transparent Generation**: As noted in our Introduction, the “black-box” nature of real-world data makes mitigating unwanted biases intractable. Simula converts this into a “white box” problem where the data composition can be explicitly designed.
> - **Auditing Tools**: The evaluation techniques we introduce in Section 2.3 can serve as safety checks. The use of taxonomies to measure data coverage can be a powerful and practical way to spot unbalanced output distributions of factors of interest. Similarly, our “calibrated attribute scoring” technique can be used to detect unwanted shifts in sensitive attributes, e.g., comparing the “level of prejudice.”

---

### Decision · Action_Editor_wQvJ · 2026-02-17

**Recommendation:** Accept as is

**Audience:**

Yes

**Audience Explanation:**

Synthetic data generation is an increasingly important topic for building AI systems. I believe the methodology, practical guidelines, and empirical results described in this paper will be of interest to the TMLR community.

**Claims And Evidence:**

Yes

**Claims Explanation:**

The paper proposes a multi-stage framework (Simula) for controllable generation of synthetic data. The framework involves an elaborate general-purpose prompting harness that includes (1) generating a taxonomy of the synthetic data given an initial prompt, (2) generating prompts to increase complexity and diversity of data and (3) filtering the resulting data.

The authors provide a detailed description of the proposed framework, and provide experiments demonstrating the effectiveness of its different components. They also show interesting results on the impact of complexity of the synthetic data, and the amount of synthetic data on the final downstream performance. Based on requests from reviewers, the authors validated their results with different optimizers (Adafactor, Muon) and models (Gemma, Qwen). Overall, the evaluation is compelling and provides sufficient evidence for the claims made by the authors.